

**Oxygen Utilization and Downward Carbon Flux in an**
**Oxygen-Depleted Eddy in the Eastern Tropical North**
**Atlantic**
**B. Fiedler[1], D. Grundle[1], F. Schütte[1], J. Karstensen[1], C.R. Löscher[1], H. Hauss[1],**
**H. Wagner[1], A. Loginova[1], R. Kiko[1], P. Silva[2], and A. Körtzinger[1,3]**
[1] GEOMAR, Helmholtz Centre for Ocean Research Kiel, Germany
[2] Instituto National de Desenvolvimento das Pescas (INDP), Cape Verde
[3] Christian Albrecht University Kiel, Germany
Correspondence to: B. Fiedler (bfiedler@geomar.de)

## 12 Abstract

The occurrence of mesoscale eddies that develop an extreme low oxygen environment at
shallow depth (about 40 to 100 m) has recently been reported for the eastern tropical North
Atlantic (ETNA). Their hydrographic structure suggests that the water mass inside the eddy is
well isolated from ambient waters supporting the development of severe near-surface oxygen
deficits. So far, hydrographic and biogeochemical characterization of these eddies was limited
to a few autonomous surveys, using moorings, underwater gliders and profiling floats. In this
study we present results from the first dedicated biogeochemical survey of one of these eddies
conducted in March 2014 near the Cape Verde Ocean Observatory (CVOO). At the time of
the survey the eddy core showed lowest oxygen concentrations of less than 5 µmol kg$^{-1}$ and a
pH of approx. 7.6 at the lower boundary of the euphotic zone. Correspondingly, the aragonite
saturation level dropped to 1 thereby creating unfavorable conditions for calcifying organisms
at this shallow depth. To our knowledge, such enhanced acidity within near-surface waters
has never been reported before for the open Atlantic Ocean. Vertical distributions of
particulate and dissolved organic matter (POM, DOM) generally show elevated
concentrations in the surface mixed layer, but particularly DOM also accumulates beneath the
oxygen minimum. Considering reference data from the upwelling region where these eddies
are formed, we determined the oxygen consumption through remineralization of organic



matter and found an enhancement of apparent oxygen utilization rates (aOUR, 0.26 µmol kg$^{-1}$
d$^{-1}$) by almost one order of magnitude when compared with typical values for the open North
Atlantic. Computed downward fluxes for particulate organic carbon (POC) at 100 m were
about 0.19 to 0.23 g C m$^{-2}$ d$^{-1}$ which clearly exceed fluxes typical for an oligotrophic open
ocean setting. The observations support the view that the oxygen depleted eddies can be
viewed as isolated, westwards propagating upwelling systems as their own.

## 1   Introduction

New technological advances in ocean observation platforms, such as profiling floats, gliders,
and in sensors have greatly facilitated our knowledge about physical, chemical and biological
processes in the oceans, and particularly those occurring on small spatio-temporal scales
(Johnson et al., 2009; Roemmich et al., 2009). In particular physical transport processes in
frontal regions and in mesoscale eddies have been found to generate biogeochemical
responses that are very different from the general background conditions (Baird et al., 2011;
Mahadevan, 2014; Stramma et al., 2013). A key process in driving the generation of
anomalies is the vertical flux of nutrients into the euphotic zone that enhances primary
productivity, a process that is of particular importance in usually oligotrophic environments
(Falkowski et al., 1991; McGillicuddy et al., 2007). Besides the locally generated response,
the westward propagation of mesoscale eddies introduce a horizontal (mainly zonal)
relocation of eddy properties. Satellite data and model studies show that eddies do play an
important role in the offshore transport of organic matter and nutrients from eastern boundary
upwelling systems (EBUS) into the open ocean. Considering their transport alone, eddies
have been found to create a negative impact on productivity in EBUS regions because of their
net nutrient export (Gruber et al., 2011; Nagai et al., 2015; Rossi et al., 2009).
The eastern tropical North Atlantic (ETNA) hosts an eastern boundary oxygen minimum zone
(OMZ) which is primarily created from sluggish ventilation (Luyten et al., 1983) and high
productivity in the EBUS along the West African coast. In its western part, the ETNA is
bounded by the Cape Verde frontal zone (CVFZ) separating the OMZ regime from the wind
driven and well ventilated North Atlantic subtropical gyre. In the south, towards the equator,
oxygen is supplied via zonal current bands (Stramma et al., 2005; Brandt et al. 2015). The
vertical oxygen distribution shows two distinct oxygen minima, an upper one at about 75m
depth and a deep OMZ core at about 400 m (Brandt et al., 2015; Karstensen et al., 2008;





Stramma et al., 2008b). On the large scale, the minimum oxygen concentrations in the ETNA
OMZ are just below 40 µmol kg$^{-1}$ (Stramma et al., 2009) but an expansion of the OMZ both
in terms of intensity and vertical extent has been observed over periods of decades (Stramma
et al., 2008a). However, recently Karstensen et al. (2015) reported the appearance of very low
oxygen concentrations at very shallow depth, close to the mixed layer base, in the ETNA in a
long term oxygen time series from a mooring at the Cape Verde Ocean Observatory (CVOO,
cvoo.geomar.de) and from a profiling float. By making use of satellite derived sea level
anomaly data, the authors could associate the occurrence of the low oxygen events with
cyclonic (CE) as well as anticyclone mode-water eddies (ACMEs). Normal anticyclones did
not show any low oxygen signature. They also propose that the oxygen minimum in CEs and
ACMEs is not being exported from the eddy formation region (along the west African coast)
but created during the westward passage of the eddies into the open ETNA.
Based on satellite data analysis a statistical assessment of mesoscale eddies has been done for
the North Atlantic in general (Chelton et al., 2011) as well as for the ETNA in particular
(Chaigneau et al., 2009; Schütte et al., 2015). However, Schütte et al. (2015 & in prep. for this
issue) were the first to further differentiate anticyclonically rotating eddies into "normal"
anticyclones and ACMEs by combining satellite data (sea level anomalies, sea surface
temperature) with in-situ data (CTD, profiling floats, glider). They found that about 1 to 2
ACMEs are generated each year (at distinct regions in the EBUS) that propagate into the open
ETNA waters.
An intense biogeochemical response in ACMEs has been reported for other ocean regions as
well. For instance, McGillicuddy et al. (2007) reported intense phytoplankton blooms in
ACMEs for the western North Atlantic, near Bermuda. They explained the phenomenon as
the result of a vertical nutrient flux driven by the interaction of the eddy with the overlying
wind field. Altabet et al. (2012) observed enhanced production of biogenic nitrogen (N$_2$)
inside an ACME in the generally suboxic conditions in the eastern South Pacific OMZ.
Further, consequences for carbon cycling such as production and export, as well as the impact
on the ETNA OMZ also remain unclear.
However, detailed process understanding of the physical and biogeochemical processes and
their linkages in eddies, in particular in the high productive ACMEs, is still scarce and one
reason is the difficulty in performing dedicated in-situ surveys of such eddies. Here, we
present the first biogeochemical insights into low-oxygen ACMEs in the ETNA based on




direct in situ sampling during two coordinated ship-based surveys. This publication is part of
a series that describe biological, chemical and physical oceanographic processes in low-
oxygen ACMEs in the ETNA. In this paper we first present the vertical hydrographic
structure of a surveyed ACME and discuss nutrients concentrations and the marine carbonate
system. All data are put into regional context by comparing ACME conditions with 1)
ambient background conditions represented by the nearby Cape Verde Ocean Observatory
time-series site (CVOO) and 2) the biogeochemical setting in the proximal EBUS off the
West African coast, were the eddy originated from. We then provide estimates for
transformation rates of various key parameters and derive estimates for carbon export rates
within the surveyed ACME in the ETNA.
## 2   Methods
Mesoscale eddies can be detected and tracked from space (Chelton et al., 2011; Schütte et al.,
2015). However, only a few of such eddies develop an oxygen depleted core and a targeted
survey of oxygen-depleted mesoscale eddies in the ETNA (and elsewhere) is challenging.
(Schütte et al., in prep. for this issue) analysed satellite and corresponding in-situ data and
found that on average about 20% of all anticyclones (10% of all eddies) are ACMEs in the
ETNA exhibit a strong low oxygen core. CEs also develop a low oxygen core but not as low
as ACMEs do.
In order to enable a targeted survey of the one particular ACME the following strategy was
designed ("Eddy Hunt" project; Körtzinger et al., introduction to this special issue): we
combined satellite data (sea level anomaly, SLA, and sea surface temperature, SST) with
Argo float data in a near-real time mode. Although we did not had access to oxygen data in
near-real time we knew from earlier observations (Karstensen et al., 2015) that low oxygen
ACMEs have a low salinity core. As such, detecting an eddy with high SLA and low SST
(note, normal anticyclones show high SST; Schütte et al., 2015) and confirming low salinity
at shallow depth from opportunistic Argo float data, potential low-oxygen ACMEs were
detected. A pre-survey (with an autonomous underwater glider) of one such candidate ACME
confirmed a low oxygen core and ship surveys were initiated.
Here, we use ship data as well as data from a profiling float of a variety of biogeochemical
parameters in order to investigate the marine carbonate system functioning on low-oxygen
eddies. The following sections will provide a brief overview of samples collected during two



ship cruises and the applied analytical methods. Moreover, the general setting of the CVOO
ship time series as well as data from hydrographic cruises and the profiling float will be
introduced.

## 2.1 Eddy Surveys

Dedicated eddy surveys were done during the RV Islandia cruise ISL_00314 (05 March – 07
March 2014; hereafter named ISL) and the RV Meteor cruise M105 (17 March to 18 March
2014; hereafter named M105). During both cruises hydrographic and biogeochemical data
was sampled in the same eddy (Figure 1). Water samples were collected with a rosette water
sampling system equipped with a CTD (conductivity, temperature & depth) and additional
sensors. Since CTD data during ISL_00314 does not meet all quality control measures
following GO-SHIP standards we expect for the hydrographic data an accuracy of about twice
the GO_SHIP standard, which is for temperature 0.002°C, for salinity 0.004 and for oxygen
sensor data approx. 4 µmol kg$^{-1}$ (note that M105 data fulfil these criteria).
Along with CTD casts, an underwater vision profiler 5 (UVP, Picheral et al., 2010) was
deployed during both cruises in order to quantify particle distribution in the water column (see
results in Hauss et al., 2015). During both cruises, CTD casts down to 600 m were performed
and trying to survey as close as possible to the eddy core (guided by the near-real time
satellite SLA maps) and likewise outside of the eddy to be able to investigate the horizontal
contrast of the eddy to the surrounding waters. Based on the SLA data the "outside stations"
during ISL and M105 were located 43 and 54 kilometres away from the supposed eddy
centre, respectively. It however tuned out that these stations where probably more at the rim
of the eddy than in the surrounding water representing typical background conditions. In order
to compare the eddy observations to the typical background conditions we used data collected
during M105 at the CVOO time series station (see section 2.2).
Additionally, a section across the eddy was performed during M105 that included multiple
hydrocasts (CTD/UVP-only, no bottle sampling) as well as current and backscatter profiles
with the ship-borne Acoustic Doppler Current Profiler (ADCP) instrument and vertically
stratified plankton net hauls (see results in Hauss et al., 2015).
For comparison, we also used data from an Argo profiling float (WMO no. 6900632) that got
trapped in a low-oxygen cyclonic eddy (Karstensen et al., 2015; Ohde et al., 2015). This float
was equipped with an oxygen sensor (AADI Aanderaa optode 3830) and a transmissometer



(CRV5, WETLabs). The given uncertainties of the float measurements were ±2.4 dbar for
pressure, ±0.002°C for temperature and ±0.01 for salinities. The float was deployed in
February 2008 at the Mauritanian shelf edge and propagated in a rather straight, west-
northwest course, into the open waters of the ETNA.
**2.2 Reference Data Sets**
Data from former research expeditions in the ETNA, conducted in other research programs
(e.g., SOPRAN, SOLAS, SFB 754), were used to put the results of the dedicated eddy
surveys into regional context. For the Mauritanian shelf area (Figure 1) three cruises were
identified that sampled the region during that part of the year when eddies are typically
created (and so was the target eddy) and released to the open Atlantic Ocean (Schütte et al.,
2015): RV Meteor cruise M68-3 (12 July – 6 August 2006) conducted a biogeochemical
survey from the Mauritanian Upwelling region up to the Cape Verde Archipelago, RV
Poseidon cruise POS399/2 (31 May – 17 June 2010) which operated in the same area and RV
Meteor cruise M107 (29 May – 03 July 2014) focused on benthic biogeochemical processes
along the Mauritanian shelf edge. Data from selected stations near the shelf edge from all
three cruises were used as a reference for biogeochemical characteristics during the eddy
formation.
Likewise, representative background conditions for the actual survey area northwest of the
Cape Verde Islands were estimated from data collected during M105 at the near-by CVOO
(17.58 °N, -24.28 °E, Figure 1). The observatory includes a ship-based sampling program and
a mooring (Fischer et al., 2015; Karstensen et al., 2015) and is located about 167 kilometres
south of the eddy survey location (at the time of the ISL sampling) in an open-ocean setting.
We used data of the CVOO sampling during M105 as background conditions in order to
illustrate local biogeochemical anomalies caused by this ACME.
**2.3 Analytical Methods**
All discrete seawater samples collected for this study were analyzed for dissolved oxygen
after Hansen (2007) with manual end-point determination. Samples were stored dark after
sampling and fixation and were analyzed within 12h on board. Regular duplicate
measurements were used to ensure high precision of measurements (ISL: 0.27 µmol kg$^{-1}$,





M105: 0.34 µmol kg$^{-1}$). Oxygen bottle data were also used to calibrate the oxygen sensors
mounted on CTD instruments.
Samples for nutrients were analyzed with autoanalyzer systems following the general method
by Hansen and Koroleff (2007). Nutrient samples during ISL and M105 surveys were always
taken as triplicates, stored at -20 °C immediately after sampling and were analyzed onshore
within 3 weeks (ISL) and 2 months (M105) after collection, respectively. Obtained precisions
from regular triplicate measurements (in µmol kg$^{-1}$) for nutrient analyses were 0.08 (nitrate),
<0.01 (nitrite), 0.02 (phosphate), 0.04 (silicate) for ISL and 0.08 (nitrate), 0.02 (nitrite), 0.05
(phosphate) and 0.07 (silicate) for M105.
Samples for dissolved inorganic carbon (DIC) and total alkalinity (TA) were preserved and
stored for later onshore analysis, following procedures recommended by Dickson et al.
(2007). Briefly, 500 mL borosilicate glass bottles were filled air bubble-free with seawater
and then poisoned with 100 µL of saturated mercuric chloride solution. Samples were stored
at room temperature in the dark and in case of later onshore analysis shipped to GEOMAR for
analysis within 3 month after sampling. Preserved samples as well as samples directly
analyzed onboard were measured using automated high precision analyzing systems
performing a coulometric titration for DIC (SOMMA, Johnson et al. 1993) and a
potentiometric titration for TA (VINDTA, Mintrop et al. 2000). High quality of obtained
results was ensured by regular measurements of certified reference material (CRM, A.
Dickson, Scripps Institution of Oceanography, La Jolla, USA; Dickson, 2010) and duplicate
samples (TA: 1.30 µmol kg$^{-1}$, DIC: 1.45 µmol kg$^{-1}$). Results from DIC and TA analysis were
used to compute the remaining parameters of the marine carbonate system (pH, $p$CO$_2$ and
$\Omega_{Ar}$) using carbonic acid dissociation constants after Mehrbach et al. (1973) as refitted by
Dickson and Millero (1987).
Samples for DOC/DON were collected into combusted (8 h, 500°C) glass ampules after
passing through combusted (5 h, 450°C) GFF filters and acidified by an addition of 80 µL of
80% phosphoric acid. The DOC was analysed with the high-temperature catalytic oxidation
method adapted after Sugimura and Suzuki (1988). Total dissolved nitrogen (TDN) was
determined simultaneously to DOC using a TNM-1 detector on Shimatzu analyser. DON
concentrations were further calculated by subtraction of measured total inorganic nitrogen
(NO$_3^-$+NO$_2^-$) from TDN. The calibrations and measurements are described in more detail in
Loginova et al. (2015) and Engel and Galgani (2015).





Filtration of seawater (1 L of seawater <150 m and 2 L >150 m depth) through a GFF filter
(0.8 µm pore size) was conducted during M105 in order to determine particulate fractions of
organic carbon and nitrogen. Filters were stored frozen (-20 °C) until analyses. In the lab,
filters were exposed to fuming hydrochloric acid to remove inorganic carbon, dried at 60°C
for ~6 hours, wrapped in tin foil and processed in an Euro EA elemental analyzer calibrated
with an acetanilide standard.
**2.4  Oxygen Utilization**
Karstensen et al. (2015) suggested that the low-oxygen cores of the eddies were created from
an enhanced respiration due to high surface productivity and subsequently sinking of
particulate matter combined with reduced oxygen supply (due to an efficient isolation of the
core from surrounding waters). The high productivity is proposed to be driven by vertical
nutrient flux into the euphotic zone, a situation that resamples coastal upwelling regions.
Therefore we compare our results of the analysis of the eddy in spring 2014 (e.g., production
and respiration of organic matter and related export fluxes) with observations from the
Mauritanian shelf.
Based on satellite SLA data the formation location of the target eddy is reconstructed to be
close to the shelf edge off Mauritania at approx. 18°N (Figure 1). This is further corroborated
by an elaborate statistical analysis of historical SLA data (Schütte et al., 2015) which
identified this region as one hotspot for the creation of anticyclonic mode water eddies
(ACMEs). We used the station data (CTD hydrocasts and discrete water sampling) from the
three cruises mentioned above which fall into the area 17.45 °N to 18.55 °N and -17.10 °E to -
16.45 °E (Figure 1). In order to account for small-scale variability of water column properties
within this area, an average profile for each investigated parameter was created by averaging
on isopycnals but mapped back to depth via the mean depth/density profile. These mean
profiles were assumed to reflect typical initial conditions of ACMEs during formation in the
Mauritanian shelf area in boreal summer (Table 1).
This reference data from the shelf was then used to determine the changes in biogeochemical
parameters en route from the formation to the survey area northwest of Cape Verde. Again,
the anomalies were determined along isopycnals and mapped back to depth. We assume that
the core of the eddy was not significantly affected by either horizontal or vertical mixing as



such ACMEs are known to host highly isolated water bodies due to their physical structure
(Karstensen et al., 2015). This assumption allows us to derive estimates for biogeochemical
rates being independent of mixing processes.
In order to determine the apparent oxygen utilization rate (aOUR) and the carbon
mineralization rate (CRR) not only the anomaly but the "age" of the eddy, that is the time
between formation on the shelf and the time the eddy surveys took place, needs to be known.
The age was determined from the SLA tracking algorithm, that was also used to determine the
area of origin (Schütte et al., in prep. for this issue; Figure 1). Biogeochemical rates were then
estimated along multiple isopycnal surfaces between the shelf and the eddy interior as shown
here for determination of CRRs:

$$CRR_i = \frac{DIC_{E,i} - DIC_{S,i}}{t_E - t_S} \qquad (1)$$

where $CRR_i$ is the carbon remineralization rate along the isopycnal surface i, $DIC_{E,i}$ the
observed DIC concentration within the eddy on isopycnal i, $DIC_{S,i}$ the average DIC
concentration on the shelf on isopycnal i, $t_E$ the time of the eddy survey, and $t_S$ the back-
calculated time the eddy was created in the shelf area. The same approach was followed to
determine rates for all other available biogeochemical variables as well.
Data from the Argo float trapped inside a CE in 2008 was processed as described in
Karstensen et al. (2015). Corresponding CRRs were derived from aOURs by applying a
Redfield stoichiometric ratio of $-O_2:C_{org} = 1.34 \pm 0.06$ (Körtzinger et al., 2001a) as no direct
measurements of the carbonate system exist for this CE.

## 2.5 Carbon Export Flux

We used CRR to estimate the shape of the vertical export flux curve for particulate organic
carbon (POC) out of the euphotic zone which is generally being described by the established
Martin Curve (Martin et al., 1987a):

$$F(z) = F_{100} \cdot \left(\frac{z}{100}\right)^{-b} \qquad (2)$$

where $F(z)$ is the POC flux at a given depth $z$, $F_{100}$ the corresponding export flux at 100 m and
$b$ a unitless fitting parameter that describes the shape of the curve.





$F_{100}$ can be determined following an approach by Jenkins (1982) using a log-linear aOUR-
depth dependence which can be also described for CRR as follows:

$$\ln(CRR) = m \cdot z + c \tag{3}$$

where m is the slope and c the intercept of the linear regression of ln(CRR) versus depth. An
estimate for $F_{100}$ can be obtained by vertically integrating $F(z)$ from 100 m downward to a
maximum depth a:

$$F_{100} = \int_{100}^{a} \ln(CRR)\, dz = \int_{100}^{a} e^{(m \cdot z + c)} dz \tag{4}$$

The *b* parameter of the Martin equation (eq. (2)) can then be determined as the slope of the
linear regression of ln(CRR) on ln(z).
The rates we derive assume that the changes can exclusively be ascribed to the
biogeochemical processes and no transport processes (ventilation) play a role. As such
reported rates in this study are to be seen as lower order estimates. However, from the
comparison of the hydrographic properties in the eddy formation area and the survey area this
assumption is plausible for the core of the eddy (see detailed discussion in section 3.1).
**3   Results & Discussion**
In the following sections, we first examine the hydrographic (section 3.1) and biogeochemical
setting (sections 3.2 - 3.4) of the surveyed ACME in a phenomenological sense. In order to
better understand and interpret the biogeochemical anomalies found in the eddy core we
compare our results with observations that are representative for either the Mauritanian shelf
region or the ambient open-ocean conditions outside of the eddy. We then derive estimates for
aOUR and carbon export rates from these data.
**3.1   Hydrography**
The Temperature-salinity (TS) characteristics of the core of ACMEs in the open ETNA
(Schütte et al., in prep. for this issue; Karstensen et al., 2015) were found to be nearly
unchanged, compared to coastal regions. They resample South Atlantic Central Water
(SACW), the predominating upper layer water mass in the Mauritanian Upwelling region.
Towards the west and north, the influence of SACW decreases and is taken over by North
Atlantic Central Waters (NACW), the dominant water mass of the ventilated part of the North



Atlantic subtropical gyre (Pastor et al., 2008). As expected for a low-oxygen eddy, the TS
characteristic in the 2014 eddy core for the two surveys matches very well with the
characteristic found from the Mauritanian shelf reference stations (Figure 2), thereby
underlining the isolation of the eddy against mixing processes with surrounding waters during
its westward propagation from the shelf into the open. However, below the eddy core
($\sigma_\theta > \sim 26.6 \triangleq \sim 250$ m) TS characteristics become more variable and no indication for isolation
is found. The upper bound of the eddy core is the mixed layer base, characterized by a very
sharp gradient (between 70 – 77 m depth) in all parameters. The vertical contrast amounts to
0.73 in salinity, 3.98°C in temperature and 165.8 µmol kg$^{-1}$ in dissolved oxygen. As expected
from the satellite analysis of Schütte et al. (2015), the mixed layer temperature were found to
differ significantly from outside-eddy conditions. Underway measurements of temperature
recorded at 5 m depth during M105 reveal colder temperatures within the eddy when
compared to outside conditions.

## 3.2  Oxygen and Nutrients

Despite quasi-constant physical water mass properties over the course of the eddy's lifetime,
changes in biogeochemical variables are observed. Continuing processes such as biological
production in the euphotic zone and organic matter respiration within the low-oxygen core as
well as underneath drive significant changes in biogeochemical properties over time. In
comparison to the reference profile from the Mauritanian Shelf we find a maximum oxygen
decrease in the eddy core (100 m) of about 57.0 µmol kg$^{-1}$ to suboxic levels (<5 µmol kg$^{-1}$;
Figure 3). We expect the oxygen decrease from continuous respiration of the organic material
that sinks out of the euphotic zone into an environment that is only minimal affected by lateral
ventilation of the eddy waters. A more detailed assessment of oxygen utilization is presented
in section 3.5.
We observe elevated nutrient concentrations (nitrate, phosphate, silicate) inside the ACME
which indicate the remineralization of organic matter (Figure 4). Nutrient data obtained
during the ISL survey showed also elevated concentrations for nitrate, nitrite and phosphate in
the mixed layer of the eddy. Such elevated surface nutrient concentrations are untypical for
the oligotrophic waters of the open ETNA but can be observed in the coastal upwelling region
(Löscher et al., 2015). As such, we expect them to be a signature of a vertical flux event. As
these elevated surface concentrations were not found during the M105 sampling we expect
that the upwelling is intermittent and/or maybe occur only locally, confined to certain regions



across the eddy. In any case, the upwelled nutrients fuel surface production, which, in turn,
draws down nutrient levels quickly again. In an oligotrophic ocean setting such an eddy with
sporadic upwelling events creates a significant anomaly when compared to ambient
conditions. Consequences on carbon cycling and sequestration are discussed in next sections
in more detail.
**3.3   Carbonate System**
By using the measured DIC and TA, the remaining two parameters of the marine carbon cycle
(pH and $p\mathrm{CO_2}$) as well as saturation levels for Aragonite ($\Omega_{Ar}$) have been calculated following
methods described in section 2.3. In accordance with the oxygen decrease already discussed, a
clear respiration signal was also found in carbon parameters (Figure 5). Values for DIC (max.
2258.8 µmol $\mathrm{kg^{-1}}$) and $p\mathrm{CO_2}$ (max. 1163.9 µatm) as well as for pH (min. 7.63) in the core of
the eddy deviate significantly from those observed in the reference profiles from the
Mauritanian Shelf region were the eddy was formed. Moreover, these values can be seen as
the highest or lowest end members for the open ETNA, respectively, thus creating an extreme
biogeochemical environment on the mesoscale. One parameter that illustrates this contrasting
environment very well is $\Omega_{Ar}$ which inside the eddy core dropped to 1.0 (i.e. the threshold
below which carbonate dissolution is thermodynamically favored; Figure 5). This value is
very much in contrast to the regional background conditions at CVOO where $\Omega_{Ar}=1$ is found
below 2500 m depth and the typical $\Omega_{Ar}$ at 100 m depth is approx. 2.4.
The horizontal gradient of pH between inside and outside eddy conditions is up to 0.3 pH
units at a water depth of approx. 100 m. It is interesting to note that a pH of 7.63 is close to
values expected for future surface ocean conditions in the year 2100 (approx. pH of 7.8) as
predicted by models assuming a global high $\mathrm{CO_2}$ emission scenario (Bopp et al., 2013).
Further, such low pH levels are used for example in artificial mesocosm experiments to
simulate these future conditions (REFERENCE!!!). Absolute values of pH inside the eddy
exceed these predictions and plankton communities inside the OMZ core are exposed to these
acidified conditions. Vertically migrating zooplankton and nekton also encounter such a
pronounced gradient during migration (see Hauss et al., 2015).
Above the core, DIC concentrations in the surface mixed layer vary between the two eddy
surveys and CVOO. Slightly higher values were found during the ISL survey when compared
to the M105 survey. The same was found for nutrient concentrations (section 3.2), which





consistently points towards a very recent or even ongoing upwelling event encountered during
the ISL sampling. Episodic upwelling within ACMEs have been reported for other regions in
the past (McGillicuddy et al., 2007).
Below the eddy core at a depth of approx. 200 m, the DIC anomaly disappears and parameters
fall back close to shelf background conditions (Figure 5). A slightly different picture is found
in profile data for TA. Here, only a minor change in TA inside the eddy core is found. This
was expected as respiration processes have a small but significant effect on TA (Wolf-
Gladrow et al., 2007). However, the major difference at depth (increased values for TA inside
the core compared to shelf background) cannot be accounted for by respiration. One potential
reason for this pattern is calcium carbonate dissolution at depth. This explanation, however,
can be excluded since both $\Omega_{Ar}$ is too high at these depths and aragonite dissolution would
also positively affect DIC concentrations (the increase of which can essentially be explained
by respiration). Thus, the more likely explanation is an intrusion of ambient NACW waters,
which, considering distinct TA-salinity relationships (Lee et al., 2006), would also affect TA
concentrations towards elevated levels. Indeed, vertical profiles for salinity (Figure 3) show
slightly higher salinity values beneath the eddy core. Furthermore, TA-salinity correlations
show different patterns when comparing between the eddy core and underneath (data not
shown) which also corroborates this interpretation.

## 3.4 Particles and Organic Matter

We used data from the UVP to illustrate vertical distribution of small particles (60 – 530 µm)
in the water column. Particle abundances show a peak at subsurface depth within the shallow
OMZ slightly below the oxygen minima observed during the ISL and M105 surveys (Figure
6). This points at accumulated particles fueling microbial respiration in the core of the eddy.
Furthermore, surface concentrations of particles significantly exceed open-ocean conditions
as found at CVOO. This is in line with Löscher et al. (2015) who described a threefold higher
primary production for surface waters inside the eddy as compared to the outside. In the
Mauritanian shelf area particle concentrations are much higher throughout the water column.
Enhanced biological production as well as influence from nepheloid layers (Fischer et al.,
2009; Ohde et al., 2015) along the shelf edge most likely cause this high level of particle
abundance. According to Hauss et al. (2015) large aggregates (>500µm equivalent spherical
diameter, UVP data) are 5-fold more abundant in the upper 600 m within the eddy than in the
usual open ocean situation in this region, suggesting a substantial increase in export flux.





Discrete bottle samples for organic carbon (POC, DOC) and nitrogen (PON, DON) were
collected during the M105 survey only (Figure 6). Both POC and DOC concentrations are
elevated inside the eddy compared to concentrations found at CVOO. In particular, POC
shows a major peak in the surface mixed layer that exceeds not only concentrations at CVOO
but also all other POC concentrations measured during the M105 cruise (including data
between Cape Verde and 7°N, data not shown). A similar picture was found for PON
concentrations. Again, these observations match very well with the findings by Löscher et al.,
(2015). Within the eddy core, only a very minor (positive) peak in POC (and PON) appears
which is located somewhat beneath the actual oxygen minimum of the core. Data below
250 m then match well with background conditions again. Vertical profiles for DOC (and
DON) also show higher values in the surface as well as a distinct (positive) peak beneath the
oxygen minimum. In contrast to the particulate fraction, DOC (DON) concentrations at depth
exceed background conditions. The position of the small POM and the pronounced DOM
peaks beneath the actual oxygen minimum is confirmed by UVP particle data (one should
note that the depth of the UVP particle peak is slightly shallower than the associated discrete
sample). The obvious minimum in DOM exactly at the oxygen minimum (Figure 6) suggests
prolonged bacterial consumption of DOM at this depth. In other words, the drawdown of
POM and DOM by bacterial respiration can be already observed right beneath the
oxycline/mixed layer base at approx. 70 m depth and intensifies towards the core of the eddy
at approx. 98 m (during the M105 survey). Below the eddy core, along with POM and DOM
peaks, an accumulation of particles with low nucleic acids content was determined (Loginova,
pers. comm.). These particles might represent ruptured or dead bacterial cells. Therefore cell
mortality could induce a release of organic matter at this depth. However, the abrupt
accumulation of particulate matter (UVP profiles, and, to a lesser extent, discrete POM data)
and DOM somewhat beneath the core remains speculative so far.
**3.5 Oxygen Utilization & Carbon Export**
Based on the differences between the observed concentrations in the eddy and the reference
profiles in the Mauritanian upwelling region the oxygen and DIC changes and respective rates
(section 2.4) were estimated (Figure 7). As outlined before, the data was compared in density
space in order to consider the large scale differences in the depth/density relation that
primarily reflects the difference in ocean dynamics (Figure 7, larger panels). As outlined in
section 2.4, the corresponding rates, presented here against depth (Figure 7, smaller panel),



were then calculated based on the estimated lifetime of the eddy (derived from satellite data).
Thus, examined rates represent mean rates over the lifetime of the eddy and do not contain
any information about their temporal evolution.
Data show clear anomalies for all parameters within the eddy core which were most
pronounced at a depth of 98 m (M105) and 105 m (ISL), respectively. Rates for all parameters
are presented in Table 1. Below the eddy core, however, rates are vanishing and become
indistinguishable from the uncertainty introduced by the applied isopycnal approach. For
instance, the assumption of a well isolated water body holds true for the core of the eddy only,
but not necessarily for deeper parts of the eddy. Here, admixture of ambient waters becomes
more likely in agreement with the TS characteristic approaching the background signature
(Figure 2), which significantly alters water mass properties of this part of the eddy. As a
consequence of the non-isolation of the water underneath the core (below approx. 250 m)
rates can not be derived using this approach and not further discussed. Similarly, rates can
also be not derived for the surface mixed layer were multiple processes modify the parameter
field (gas, heat and freshwater exchange).
The apparent oxygen utilization rate (aOUR) within the eddy peaks at 0.26 $\mu$mol kg$^{-1}$ d$^{-1}$
(M105 survey) in the oxygen minimum which corresponds to the $\sigma_\theta = 26.35$ isopycnal. This
aOUR is one of the highest values which have been reported so far for the ETNA. Karstensen
et al. (2008) derived large scale thermocline aOUR from transient tracer data and AOU values
and found a mean aOUR of 0.03 $\mu$mol kg$^{-1}$ d$^{-1}$ in the similar depth range (similar to other
estimates such as Jenkins 1982). However, from a low-oxygen CE a direct estimate based on
an Argo float that was trapped in an eddy revealed 3 to 5 times higher rates (Karstensen et al.,
2015). In the same study, an aOUR of 0.25 $\mu$mol kg$^{-1}$ d$^{-1}$ within another ACME was found
based on an approach similar to ours by comparing oxygen in the upwelling region with the
oxygen concentrations 7 months later. The smaller rates found in the cyclonic eddy might
indicate a less isolated core but could also be related to the steady mixed layer deepening in
the CE which may provide a diapycnal oxygen pathway. However, in summary aOUR within
CEs as well as ACMEs significantly exceed typical rates in the ETNA.
Rate estimates for other biogeochemical parameters within the investigated ACME are also
exceptionally high (Table 1). We compared estimated rates with each other by looking at
stoichiometric ratios such as C:N, N:P and –O:C (data not shown). In fact, all ratios were
found to be close to, or not distinguishable from, the stoichiometry proposed by Redfield et al.





(1963). This finding provides indication for a reliable assessment of biogeochemical rates
based on the assumptions that were made and on independent samples of multiple parameters
taken during two independent cruises.
The observed DIC increase rate within the eddy core can be referred to as the CRR resulting
from continued respiration of organic matter. As illustrated in Figure 5, the peak in DIC
coincides with the depth of the sharpest decrease of POM and DOM. This is to be expected,
as the CRR should equal the derivative of the vertical POC flux curve with respect to the
depth. Following the approach of Jenkins (1982) one can derive the vertical flux of POC from
aOUR or CRR values, respectively. Downward fluxes for POC can be seen as the major
export process of carbon out of the euphotic zone.
We used these CRRs within the eddy core for determination of the vertical POC flux at
different depths by means of a power law function (Martin et al., 1987b). Vertical integration
of the data between 100 m and 1000 m yielded estimates of the vertical POC flux at 100 m
during the ISL and M105 cruises of 0.19 (± 0.08) and 0.23 (± 0.15) g C m$^{-2}$ d$^{-1}$, respectively
(Figure 8). These values are exceptionally high both for the ETNA but also for other open-
ocean regions. Table 2 provides a brief overview of studies that determined POC fluxes at
different locations based on different methods. In the open ETNA, recently determined POC
fluxes at 100 m from floating sediment trap deployments (Wagner et al., pers. comm.) were
lower by a factor of approx. 3 than inside the ACME. Interestingly, the same authors revealed
POC fluxes at the Mauritanian shelf edge in the same magnitude as found inside the
investigated ACME. This supports the view that these ACMEs can be viewed as isolated,
westwards propagating upwelling systems as their own.
POC fluxes derived here generally show higher values than found in other open-ocean studies
but are comparable to values associated with a North Atlantic spring bloom event (Berelson,
2001). Moreover, POC fluxes for this ACME were also in line with estimates made for other
eddies, such as enhanced POC fluxes determined at the rim of a CE in the Western Pacific
(Shih et al., 2015) or inside a CE in the ETNA (Figure 8, derived from aOUR data in
Karstensen et al., 2015). In general, estimated POC fluxes for the surveyed ACME based on
the method described in section 2.5 may represent a rather conservative estimate as the aOUR
was derived based on the assumption of complete absence of vertical and horizontal
ventilation processes. Thus, any minor ventilation process affecting the eddy core would
cause our OURs and POC flux estimates to be biased low.





The corresponding *b* parameter of the Martin curve for the two ACME surveys are high (1.55
– 1.64, Figure 8) when compared with typical open-ocean values. High *b* values indicate steep
and thus local flux attenuation in the upper layer which, in our case, could be explained by the
vertical structure of the ACME with its well-isolated local core. Again, our findings for flux
attenuation are comparable to those obtained during a North Atlantic bloom experiment
(Berelson, 2001) but also to observations recently conducted in the North Atlantic subtropical
gyre (Marsay et al., 2015). Controversial discussions in the scientific literature exist about
different dependencies of the *b* parameter. For instance, Marsay et al. (2015) also compared
POC flux determinations from four different sites in the North Atlantic with each other. They
found a positive correlation between water temperature and the *b* parameter in the North
Atlantic. Berelson (2001) proposed a linear relationship between the POC flux at 100 m and
the *b* parameter which also matches with our data. In contrast, a few studies also suggest a
dependency between the *b*-parameter and ambient oxygen concentrations with lower *b*-values
found in low oxygen environments (Devol and Hartnett, 2001; Van Mooy et al., 2002). Our
data do not reflect this relationship, most likely due to physical processes inside the eddy such
as local upwelling and redistribution of particulate matter which may alter the shape of the
downward POC flux. Since we are lacking direct flux measurements and only have a very
limited number of observations we are not able to appropriately de-convolve drivers of the
derived POC flux attenuation profile inside this ACME.
**4   Conclusions**
We performed two biogeochemical surveys within an ACME in the open ETNA off West
Africa near the CVOO time-series site. The core of this mesoscale eddy was found to host an
extreme biogeochemical environment just beneath the surface mixed layer. The concentration
of oxygen had dropped to suboxic levels as a consequence of severely hindered vertical and
horizontal ventilation of the core along with continuing remineralization during the eddy's
lifetime. There is evidence that moderately elevated nutrient concentrations in the top layer of
the ACME are caused by (episodic) upwelling events and fuel an enhanced surface primary
productivity that moves with the ACME. Likewise, nutrient concentrations as well as $p$CO$_2$
levels showed an intense increase which created significant anomalies when compared to
ambient open-ocean ETNA conditions. Values of pH, for instance, indicate highly acidified
waters at the lower edge of the euphotic zone which corresponds to $\Omega_{Ar}$ values of 1.





We also investigated magnitudes of biogeochemical processes occurring within the eddy
during its westward propagation such as apparent oxygen utilization and carbon
remineralization by comparing our survey data with conditions prevailing during the ACME's
initial state (Mauritanian shelf). Results showed mean aOURs over the lifetime of the ACME
that exceed typical rates in the open-ocean ETNA by an order of magnitude (Karstensen et al.,
2008). Resulting POC fluxes inside the ACME was also found to exceed background fluxes
in the oligotrophic ETNA by a factor of two to three and thus are comparable to meso- and
eutrophic regions such as the Mauritanian upwelling region or the subpolar North Atlantic
spring bloom. This finding is also in line with a three-fold enhanced primary productivity in
the same ACME's surface layer derived from Löscher et al. (2015) based on seawater
incubations. Our results confirm that ACMEs in the ETNA can be seen open-ocean outposts
that clearly exhibit their origin in the EBUS but through their continued biogeochemical
activity at the same time represent alien biogeochemical environments in a subtropical ocean
setting.
The results of this study, however, are based on two independent surveys carried out at a
certain point of time in the lifetime of the ACME. Thus, we are not able to address questions
about the evolution and (non-) linearity of processes within the ACME throughout its lifetime.
Therefore, future surveys should resolve not only spatial structure but also temporal evolution
of biogeochemical processes at different life stages of these eddies.
In addition to this biogeochemical investigation, two other studies have documented the
impacts of this low-oxygen ACME on zooplankton and microbial communities (Hauss et al.,
2015; Löscher et al., 2015). There is empirical indication that future scenarios such as
deoxygenation and ocean acidification can also affect higher trophic species (Munday et al.,
2010; Stramma et al., 2012). Any possible influence of this ACME on higher trophic levels,
however, remains unknown and would require a different observational approach. The
discovered anomalies within this eddy can be seen as a large (50-100 km diameter) and
relatively long-lived (~1 year) mesocosm featuring the development of low-oxygen and low-
pH conditions in a completely unmanipulated natural environment. Hence, investigating the
full range of this mesocosm-ecosystem will provide useful data and may help to better
understand ecosystem responses to future ocean conditions.





**Acknowledgements**
The authors would like to thank Meteor M105 chief scientists M. Visbeck and T. Tanhua for
their spontaneous support of the "Eddy Hunt" project, as well as H. Bange and S. Sommer for
providing hydrographic data for the Mauritanian shelf area. Conducting field work at Cape
Verde would not have been possible without the tremendous support and engagement of the
CVOO team at INDP (Ivanice Monteiro, Nuno Vieira and Carlos Santos) as well as S.
Christiansen and T. Hahn. For DIC, TA, nutrient and DOC/TDN sample analysis we thank S.
Fessler, M. Lohmann and J. Roa. Processing of CTD data was performed by G. Krahmann
and S. Milinski. We also appreciate professional support from captains and crews of RV
Islândia and RV Meteor.
This study received financial support from the Cluster of Excellence "Future Ocean" (grant
no. CP1341, "Eddy Hunt"), the BMBF project SOPRAN (grant no. 03F0662A), the DFG
Collaborative Research Centre 754 and the European Commission for FP6 and FP7 projects
CARBOOCEAN (264879) and CARBOCHANGE (264879).



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





Table 1     Overview of detected concentration anomalies ($\Delta_{total}$) within the ACME core
during the two surveys referenced against prevailing conditions at the shelf. Rate estimates
are based on the lifetime of the ACME derived from satellite sea level anomaly data (ISL: 163
days, M105: 173 days). Values for the average shelf profile are given in order to illustrates
local variability at the corresponding isopycnal (=26.35 kg m$^{-3}$ - 1000).

| | ISL | | M105 | | Shelf | |
|---|---|---|---|---|---|---|
| | 05 – 07 March 14 | | 17-18 March 14 | | June / July | |
| | $\Delta_{total}$ (unit) | Rate (unit d$^{-1}$) | $\Delta_{total}$ (unit) | Rate (unit d$^{-1}$) | Mean (unit) | SD (unit) |
| Salinity (psu) | -0.082 | < 0.004 | -0.054 | < 0.002 | 35.588 | 0.124 |
| Temp. (℃) | -0.280 | -0.002 | -0.184 | -0.001 | 15.353 | 0.415 |
| $O_2$ (µmol kg$^{-1}$) | -35.56 | -0.22 | -44.42 | -0.26 | 48.95 | 8.88 |
| $NO_3^-$ (µmol kg$^{-1}$) | 3.48 | 0.02 | 5.02 | 0.03 | 25.77 | 1.62 |
| $NO_2^-$ (µmol kg$^{-1}$) | -0.08 | < -0.001 | < -0.01 | < 0.001 | 0.09 | 0.11 |
| $PO_4^{3-}$ (µmol kg$^{-1}$) | 0.29 | < 0.01 | 0.34 | < 0.01 | 1.60 | 0.14 |
| $SIO_2$ (µmol kg$^{-1}$) | 2.05 | 0.01 | 2.52 | 0.01 | 6.73 | 1.27 |
| DIC (µmol kg$^{-1}$) | 35.1 | 0.2 | 39.8 | 0.2 | 2218.7 | 1.4 |
| TA (µmol kg$^{-1}$) | -10.8 | < 0.1 | -12.3 | < 0.1 | 2331.5 | 7.5 |
| $p$CO$_2$ µatm | 268.68 | 1.65 | 332.67 | 1.92 | 827.93 | 28.15 |
| pH | -0.12 | < -0.01 | -0.14 | < -0.01 | 7.77 | 0.01 |
| $\Omega_{Ar}$ | -0.38 | < -0.01 | -0.43 | < -0.01 | 1.48 | 0.08 |




1 Table 2   Comparison of $F_{100}$ values from the literature representing different ocean

2 regions with the results of this study.

| Region | $F_{100}$ (g C m$^{-2}$ d$^{-1}$) | Method | Reference |
|---|---|---|---|
| ETNA (ACME) | 0.19 – 0.23 | aOUR | this study |
| ETNA (CE) | 0.24 | aOUR | this study (data from Karstensen et al. 2015) |
| West Pacific (CE) | 0.13 – 0.19 | Trap | Shih et al. 2015 |
| ETNA (open ocean) | 0.11 | aOUR | Karstensen et al. 2008 |
| N. Atl. (bloom) | 0.29 | Thorium, Trap | Berelson 2001 |
| Arab. Sea | 0.03 – 0.11 | Thorium | Lee et al. 1998 |
| N. Pac. Gyre (HOT) | 0.03 | Trap | Buesseler et al. 2007 |
| N. Pac. (K2) | 0.03 – 0.08 | Trap | Buesseler et al. 2007 |
| N. Atl. (Gyre) | 0.02 | Trap | Marsay et al. 2015 |
| N. Atl. (Gyre) | 0.15 | aOUR | Jenkins 1982 |
| NE Pac. | 0.05 | Trap | Martin et al. 1987b |

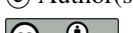



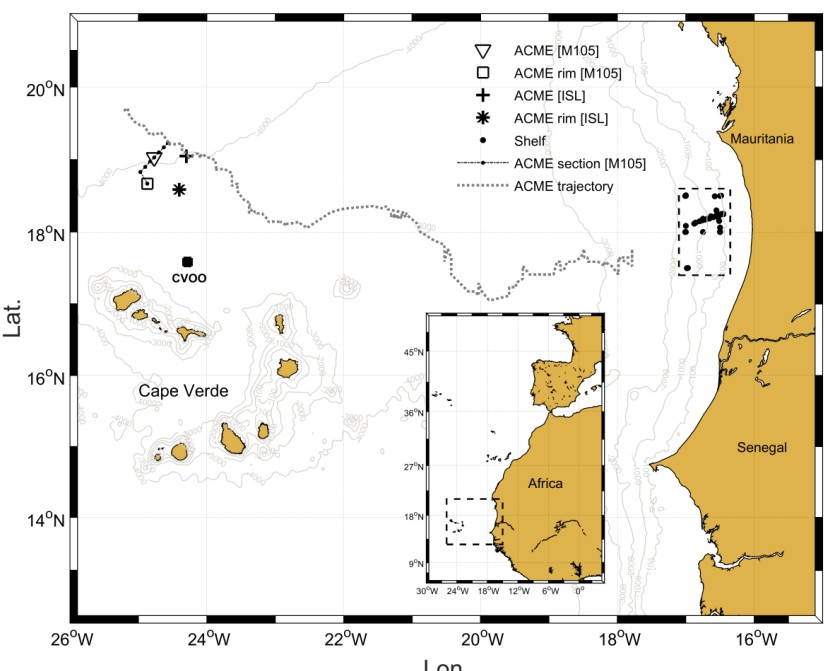

Figure 1 Map of the study area between the Mauritanian coast and the Cape Verde
Archipelago. The ACME trajectory (dotted line) is based on satellite sea level anomaly data
and starts off the Mauritanian shelf edge in Sept. 2013. In March 2014, the ACME was
surveyed twice north of Cape Verde with two different research vessels: RV Islândia (ISL)
and RV Meteor (M105). The area marked on the Mauritanian shelf (dashed line) represents
the area where the ACME was most likely created and which serves as a reference for initial
conditions within the eddy.





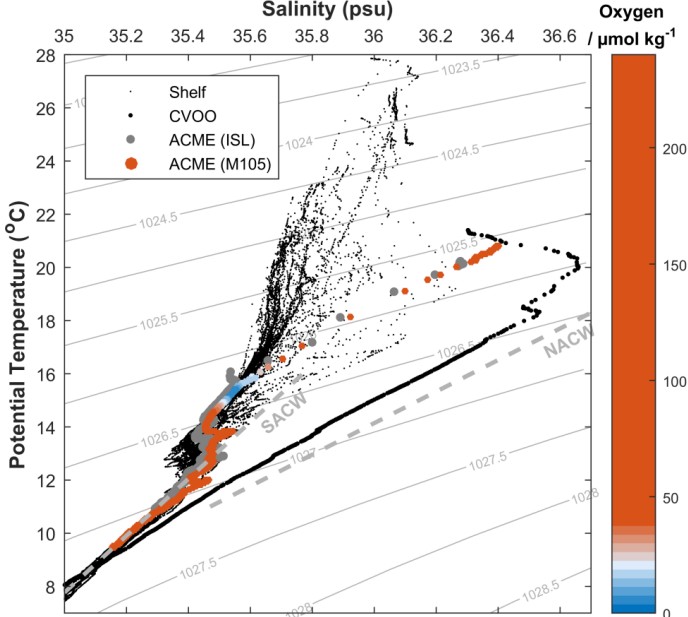

Figure 2      Temperature-Salinity (TS) diagram containing data from both eddy surveys
(colored and gray dots), the nearby CVOO station (large black dots) and accumulated CTD
hydrocast data from multiple surveys on the shelf (small black dots). Dashed gray lines
indicate typical NACW and SACW water mass signatures after Tomczak (1981).





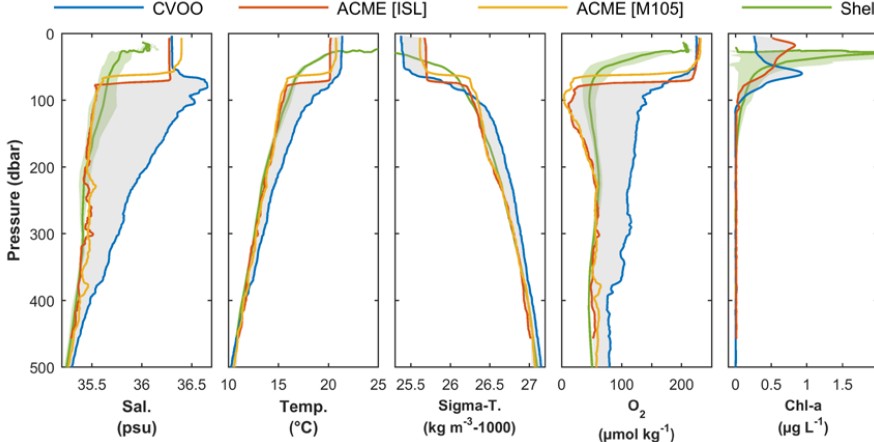

Figure 3 Vertical profiles for all parameters measured from sensors mounted on CTD rosette
systems. Data from the nearby CVOO station (blue) represent local background conditions,
the gray area emphasizes the local anomaly against the background introduced by the ACME
(yellow and red) and the green curve represents mean initial conditions of the ACME at the
shelf (light green indicates standard deviation of the mean profile). Note that not all surveys
were carried out with the same sensor package.



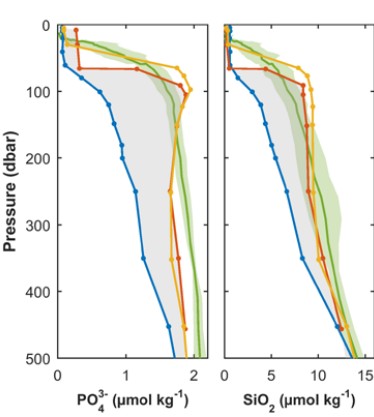

3   Figure 4        Discrete bottle data for nutrients from the different ACME surveys. The grey

4   shading illustrates the anomaly of the ACME (ISL) with respect to the regional background

5   situation (CVOO).





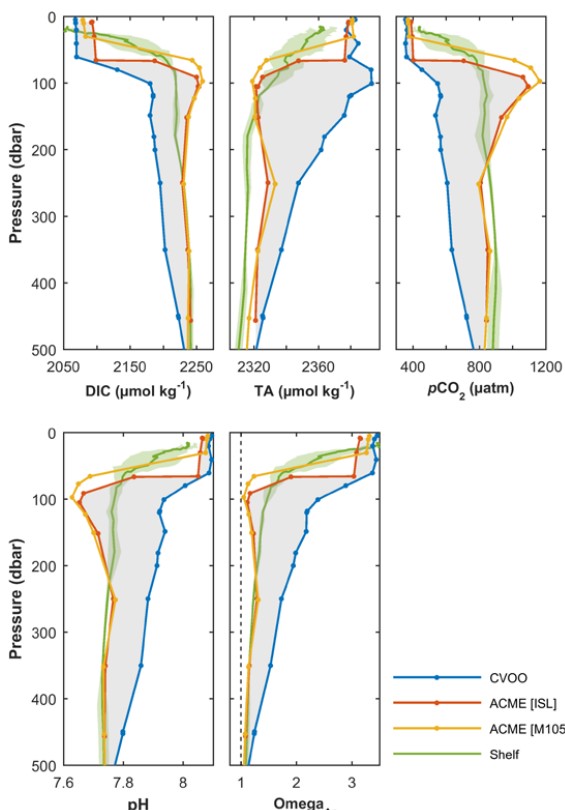

2      Figure 5        Discrete bottle data for DIC and TA and calculated parameters of the carbonate

3      system (pH, $p$CO$_2$ and $\Omega_{Ar}$) from the different ACME surveys. The grey shading illustrates

4      the anomaly of the ACME (ISL) with respect to the regional background situation (CVOO).



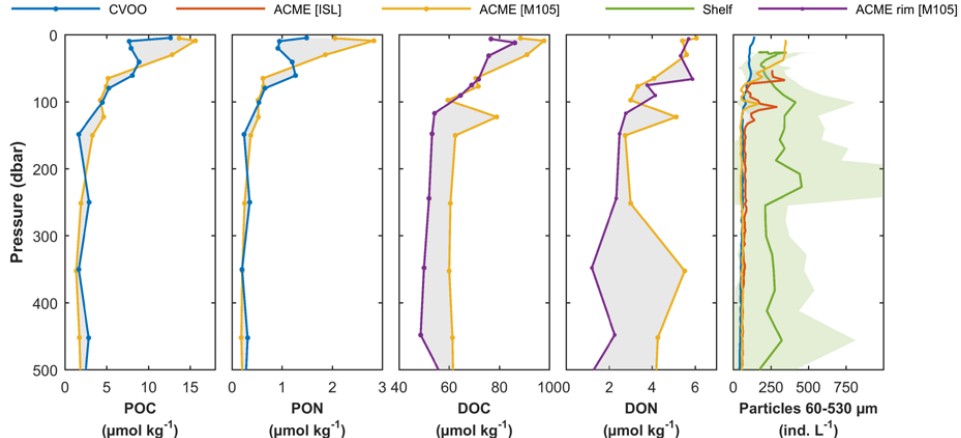

Figure 6   Vertical Distribution of particulate and dissolved organic matter (first 4 panels)
based on discrete samples and particle density (60 – 530 µm) derived from high resolution
UVP data (right panel). Note that no data at CVOO exist for DOC and DON, hence data from
the eddy rim station is shown.

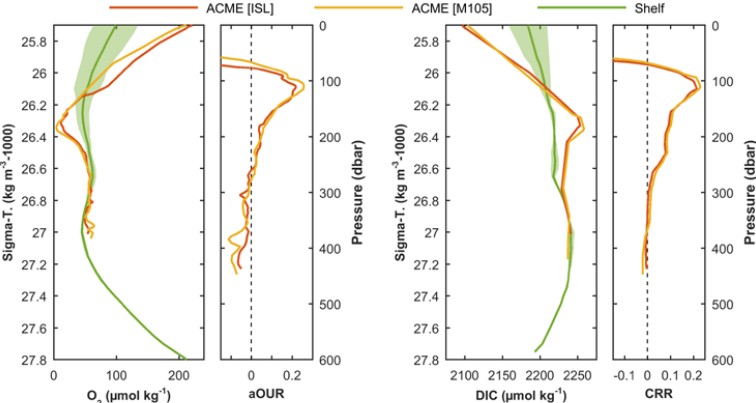

Figure 7   Estimated biogeochemical rates within the ACME as derived along isopycnals
between the shelf (green) and the ACME at the time of the two surveys (red, yellow). This
approach is illustrated for oxygen and DIC profile data (large panels). Corresponding aOUR
and CRR are peaking in the core of the ACME (small panels). Note that the matching
between shelf and ACME data was made in density space whereas the resulting rates are
plotted in depth space.



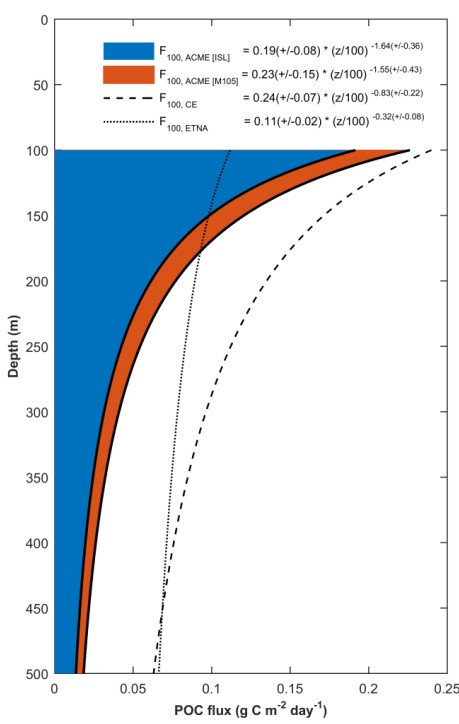

Figure 8    Derived downward POC fluxes based on a model after Martin et al. (1987b) for the two ACME surveys (blue and red), a cyclonic eddy sampled by an Argo float (CE, dashed line; Karstensen et al., 2015) and the general ETNA (Karstensen et al., 2008). Flux estimates for the two ACME surveys are based on CRRs estimated from DIC sample data. For the CE, aOURs derived from oxygen measurements on an Argo float were converted to CRRs by applying a stoichiometric $-O_2$:C ratio of 1.34 (Körtzinger et al., 2001b). Background POC flux in the ETNA was estimated from large scale thermocline aOURs derived from transient tracer data and AOU (Karstensen et al., 2008) followed by a stoichiometric conversion as described above.