# Peer review of "Oxygen Utilization and Downward Carbon Flux in an"

_Biogeosciences, 2016_

## Referee Comment (RC1) · Anonymous Referee #1 · 9 Mar 2016

Review: Oxygen Utilization and Downward Carbon Flux in an Oxygen-Depleted Eddy in the Eastern Tropical North Atlantic

Author(s): B. Fiedler et al.

MS No.: bg-2016-23

General Comments:

This is a descriptive study presenting observations of physical and biogeochemical properties of an anticyclonic mode water eddy (ACME) in the eastern tropical North Atlantic (ETNA). The topic is a current one and studies addressing biogeochemical characterization of mesoscale eddies are currently limited. The observational approach

using dedicated survey is novel, and presents a framework for the design of future studies. The main result is intriguing, suggesting that physical properties are preserved within the core of the eddy, while biogeochemical properties are evolving with time as the eddy 'ages' (the consequence being severe oxygen undersaturation and low pH and aragonite saturation state). As such, these systems could serve as a natural mesocosm for studying ecosystem responses to severe ocean conditions in the future, e.g. OA. I feel this study therefore makes significant contribution to our understanding of biogeochemical processes in low oxygen ACME. I recommend the paper to be accepted with minor revisions. Below are my comments and suggestions:

General comments:

1) Boundary definition: The authors are rather vague about the definition of eddy boundaries (the lateral boundaries, but in particular the lower boundary of the eddy core), as well as the processes leading to exchanges, or lack thereof, across those boundaries. I would suggest the authors provide more details on eddy boundaries, the depth of the mixed layer, and the depth of euphotic zone, as well as provide a stronger case for rationalizing why eddy waters don't mix with the surrounding ocean waters. For example, is the depth of the euphotic zone (Dez) and mixed layer depth in the eddy core equal to the outside water? Or is Dez different due to higher light attenuation by particles?

2) Episodic events: Throughout the paper a steady state biogeochemical system is implied (or at least a slowly evolving biogeochemical state). Yet the physical processes that allow for these balances are episodic and submesoscale (e.g. evidence for the re-supply of nutrients into the upper layer is lacking, and yet required for the equilibrium biogeochemical state in the mixed layer). Is this vertical nutrient flux driven by interaction of eddies with the overlying wind field, Ekman pumping, or internal waves displacing isopycnals and mixing? How frequent are these episodic events?

3) Downward carbon flux - POC export: This seem to be the weakest part of the paper.

The authors need to provide more details on physical and biological assumptions in the simple downward flux model, and whether its assumptions are valid in the eddy core. Is a steady state balance implied for model? A constant diffusivity? What about small non sinking POC export by eddy flow field subduction of surface waters with high POC concentrations? (See Mahadevan et al., 2015, Science). Consequently, the carbon flux model and Figure 8 may not make a meaningful contribution to the study.

4) Significance (and a Budget): What is significance of 1 to 2 ACMEs generated every year that propagate into the open ETNA waters for biogeochemistry and ocean acidification of the region, or even the ocean basin? Is this a phenomenon that might be expect to occur elsewhere (does the study have much broader implications? To help address this it might be worth carrying out at a budget/balance exercise to quantify, for example, whether the supply of nutrients exceeds export of nutrients, which leads to than increased productivity.

Detailed comments:

Abstract:

1) Page 1, Line 13: Define the extreme low oxygen environment.

2) Page 1, Line 22: Define the lower boundary of the euphotic zone.

3) Page 1, Line 27: Define the lower boundary of the surface mixed layer. Is this shallower than the euphotic zone?

4) Page 2, Line 1: an enhancement of apparent oxygen utilization rates...at what depth?

5) What is the significance of your findings for the biogeochemistry of ETNA?

Introduction:

6) Page 2, Paragraph 2: A figure showing ETNA, OMZ, EBUS, , CVFZ would be helpful. Maybe add on to Figure 1.

7) Page 3, Line 9: Define anticyclone mode-water eddies.

8) What is the main objective of this paper?

Methods:

9) Page 5, Line 8: At what depth were water samples collected?

10) Page 6, Line 26: At what depth were the water samples for DO collected?

11) Page 7, Line 10: same as above, define depth of DIC and TA samples.

12) Page 7, Lines 21-24, using which software?

13) Page 9, Line 4: Define apparent oxygen utilization rate.

Results and Discussion:

14) Page 10, Lines 15-20: This text is repetitious, and can be omitted.

15) Page 10, Lines 22-24: Specify depth.

16) Page 11, Line 20: (100m)- not clear if is this at 100m, or from 0 to100m?

17) Page 11, Lines 16-18 (sentence 2): move to after describing the results.

18) Page 11, Line 27: Why elevated nitrate, nitrite and phosphate but not silicate? Also, did you look into nitrate:phosphate ratio as evidence of denitrifying bacteria?

19) Page 12, Lines 1-5: How often do these sporadic events occur? Are they wind induced, or due to passage of internal waves?

20) Page 12, Lines 7-9: This sentence is a repetition from Methods and can be omitted.

21) Page 12, Line 26: "phytoplankton communities are exposed to these acidified conditions". How often? (the pH minimum is located just below the euphotic zone).

22) Page 13, Lines 25-29: is this consistent with your Chl-a observations?

Conclusions:

23) Page 17, Line 30: "intense increase"...specify where.

24) Page 18, Line 9: Is the 3-fold increase in primary productivity consistent with your observed Chl-a?

25) Page 18, Lines 20-25: I would replace this text with a sentence on the significance of the 1 to 2 ACMEs generated every year that propagate into the open ETNA waters for biogeochemistry and ocean acidification of the area.

Tables:

26) Table 1: Add: negative values correspond to... Are these average anomalies over some depth range?

Figures:

27) Figure 2: SACW missing?

28) Figure 8: This figure and the associated carbon flux model (eqn 2) does not make a significant contribution to the paper as it stands. See general comment #3 above.

---

## Referee Comment (RC2) · Anonymous Referee #2 · 19 Mar 2016

**Summary**

The paper by Fiedler et al. describes a mode water eddy surveyed in the Eastern Tropical North Atlantic and establishes a link between the local biological activity and the extremely low oxygen concentration in the eddy core. To do so, Fiedler et al. present and analyze the data collected by two research vessels in correspondence of an anticyclonic mode water eddy detected in March 2014 in the region above Cape Verde Islands. They compare these measurements with both data referring to the ambient background condition in the Tropical North Atlantic and data collected on the Mauritanian shelf where the eddy was formed. The mode water eddy shows

sub-oxic conditions in the core at about 100m depth. Fiedler et al. demonstrate that the extremely low oxygen concentration is associated with a decrease in pH, a drop in the aragonite saturation level, and a higher DIC and higher nutrient concentration respect to the open Atlantic measurements. They find that the low-oxygen core is characterized by a drop in particle concentration, POM and DOM respect to the highly-productive surface of the eddy. The authors calculate a remarkably high apparent oxygen utilization rate and POC flux for the eddy core, using as a reference the data collected from the Mauritanian shelf. All these results are interpreted as a clear sign of elevated respiration rates in the oxygen-depleted eddy center, and support the causal connection between such biological activity and the remarkably low oxygen concentration.

**Contribution**

Oxygen-depleted mode water eddies have first been described and studied only a few years ago, and constitute a very novel and interesting object of study. Many mechanisms regarding the formation of this extremely low-oxygen environment still need to be clarified. The connection between physical and biological processes during the evolution of these kind of eddies is still unclear and the biogeochemical data are still scarce. Oxygen-depleted mode water eddies represent some interesting natural laboratories for the study of biological activity in suboxic conditions, and are relevant for the prediction of extreme biological conditions. In fact, they can release their oxygen-depleted water when crossing an obstacle such an island, drastically impacting the local ecosystem and representing a sort of wandering bomb.

Fiedler et al. present some new and exciting biogeochemical data collected in correspondence to such an eddy, contributing to clarify the characteristics oxygen-depleted eddies with some precious piece of information. They present an interesting

comparison with the open Atlantic background state that supports the hypothesis that the oxygen-depletion is due to intense local biological activity that strongly contrasts with the surrounding oligotrophic waters. They highlight the implications that the decrease in pH and the Aragonite saturation in the eddy core level may have on the local and potentially impacted ecosystems. However, they are not able to shed light on the evolution of this biological activity during the eddy lifetime (about one year) due to the scarcity of data collected by the two cruises.

Fiedler et al. also calculate the rates of apparent oxygen consumption and POC fluxes in the eddy core and find very high values for both quantities compared to the expected values for an oligotrophic region. These results strengthen the hypothesis in which the oxygen depletion is a consequence of high respiration in the eddy core. For these calculations, they use the data collected by 3 independent cruises on the Mauritanian shelf as a reference for the initial biogeochemical composition of the eddy. These cruises were conducted in 3 different years in the shelf region that is statistically the most likely region of eddy formation. The authors' calculations rely on the assumption of representativeness of these shelf measurements and on the assumption of complete isolation of the eddy core from the surrounding environment, from the eddy formation region on the shelf to the offshore region where it was sampled.

**Recommendation**

I find the subject of the paper novel, interesting and well suited for being published in Biogeosciences. The results of this paper are important for making progress towards the understanding of the biogeochemical activity in anticyclonic mode water eddies and explain the processes behind the formation of their oxygen-depleted core. However I think that the authors need to better motivate the assumptions

made for the POC flux and apparent oxygen consumption calculations (see "Major comments", nr.1). Moreover, aside from the scientific aspects of the paper, I feel the urge to underline the necessity to improve and adjust the style of the writing and to proof-read the paper for typos and syntax. Some of the basic information in the paper needs to be reorganized and clarified in a more efficient way to ease the communication of the scientific content. At the present state, the paper is extremely difficult to read and contains several major oversights that make it sound like a draft paper. In my opinion, an improvement of this aspect is essential for making the paper publishable. I suggest that this paper is accepted after major revisions are made.

**Major Comments**

**1) Reference Data Set: an assumption that needs to be strengthened**
The three Reference Data Sets used by the authors for the bgc fluxes calculations are from cruises that were conducted on the Mauritanian shelf around the second-half of July 2006, beginning of June 2010, and beginning of June 2014. The surveyed eddy is supposed to have formed on the Mauritanian shelf around June/July 2013. This can be inferred from the paper, but it's not explicitly stated. At page 8 lines 17-21 the authors explain how they reconstructed the region of origin of the eddy on the shelf on the base of statistical analysis of historical SLA, and how this region coincides with the location of the 3 Reference Data surveys. However, the eddy trajectory from SLA in "Figure 1" starts about two months later (Sept. 2013) at least 100km in the off-shore direction. The fact that the trajectory of this specific eddy was not retrieved on the shelf that may imply, for example, that the eddy boundaries of this eddy were not already well formed, therefore the eddy may have continued to trap water while leaving the shelf area of the Reference Data Sets, or may have been spun by a lateral filament.
At page 11 lines 4-5 the authors underline the matching between the Temperature and Salinity of the eddy core when it was surveyed and the reference station measurements on the shelf. This supports the assumption of isolation of the eddy core from the shelf to the offshore waters. However, biogeochemical properties can be more variable than physical properties on both spatial and time scales, especially in active shelf regions. The authors write about the Reference Data Sets [page 8 lines 23-27] "in order to account for small scale variability [. . .] an average profile for each investigated parameter was created [. . .]. These mean profiles were assumed to represent typical initial conditions of ACMEs [. . .]".

Given the complex dynamics of the flow around the shelf edge, the time and spatial variability of biogeochemical processes, the timescale of sporadic upwelling events; given the fact that the eddy trajectory from SLA was not retrieved in the shelf region, and the complexity of the eddy formation process:

1. Can the choice of these Reference Data Sets be better justified? Are there no available data for the region in which the trajectory was actually retrieved in Sept.2013?

2. How do mean profiles account for small scale variability?

3. Is it possible to exclude strong discontinuities (input of external water, sediment resuspension, interaction with other forming eddies, etc.) in the eddy evolution between the shelf region of the Reference Data Sets and beginning of the track in "Figure 1"?

4. Several times in the article the authors refer to sporadic upwelling events fueling the high surface productivity in the eddy. How is the hypothesis of "production being boosted in the surface of the eddy by upwelling events" compatible with the hypothesis of "complete isolation of the eddy core" along the whole eddy lifetime? What is the spatial distribution of these upwelling events in the eddy?

**2) Description of the surveyed eddy**

The authors do not provide a clear general description of the characteristics of the surveyed eddy, among which some basic details: date (month/year) and coordinates of the eddy when forming on the shelf; date and coordinates of the beginning of the

track; date, coordinates, radius, shape and age of the eddy when surveyed. Some of these characteristics can be retrieved in different parts of the article explicitly or implicitly, but it is the work of the reader to collect them. I suggest presenting these characteristics in a dedicated paragraph where the eddy is introduced and described. As regard to "Figure 1": it may be helpful to add a timescale of the eddy trajectory and to draw the eddy contours in the region of the cruises, to understand where the measurements were located with respect to the eddy center and boundaries.

Most of the observations described in the chapter "3 Results  Discussion" would be much easier to understand if a nice description of the vertical physical structure of the surveyed eddy was given.

**3) Description of the dedicated eddy surveys**

In paragraph "2.1 Eddy surveys" the number of samples that were collected during each cruise is not clear and some of the descriptions are confusing. As the 2 cruises are described as "first dedicated biogeochemical surveys" of the eddy the reader may expect to see some 2D biogeochemical sections and wonder what the spatial resolution of the samples is. However, it becomes clear later that the biogeochemical data analyzed in the article consist in only 2 bottle measurements per cruise, 1 per cruise referring to the eddy center. I suggest stating this clearly in the text.

The CTD/UVP-only (no bgc) section M105 is introduced but never plotted or clearly discussed. At page 5 lines 19-24 the authors write that some stations supposed to be at a certain distance from the center of the eddy on the base of SLA "turned out probably more at the rim of the eddy than in the surrounding water representing typical background conditions". What is the reason for this conclusion? Can this be elaborated more in depth to justify this sentence?

**4) Quantitative results**

Some of the results presented in the sections from 3.1 to 3.4 are not well quantified. Data collected in the eddy center and data on the shelf or in the open Atlantic are often compared with not-well-defined or confusing terms. I strongly suggest giving to the descriptions a more quantitative flavor.

**5) Style: English, typos**

The paper, apart for a few sections, is scattered with typos, misspellings and incorrect formulation of the English sentences. Sentences are often very convoluted and difficult to follow. The frequent use of bracketed subordinates makes the reading process even more complicated.

I highly suggest a proof-reading of the paper for typos, grammar and syntax, as well as a simplification of the structure of the sentences and the limitation of the use of brackets to the very essential. Some errors are listed in the "Detailed comments" section.

**Detailed comments**

1) Introduction: The section "Introduction" of the paper ends with a paragraph (from page 3 line 29, to page 4 line 10) that introduces the content of the article. However, it forgets to anticipate any section about consequences and conclusions of the present study. I suggest to strengthen this paragraph in this sense, anticipating to the reader the presence of relevant conclusions connected to the results.

2) page 2 line 25: Eastern Tropical should be capitalized when defining acronym

3) page 3 lines 18-20: incorrect syntax, "that" should follow the name that it refers to (ACME)

4) page 3 line 27: commas out of place

5) page 3 line 29: word "process" is redundant

6) page 4 line 2: misspelling "describes"

7) page 4 lines 16-19: sentence beginning with brackets; confusing sentence, "in the ETNA" better after "in situ-data", maybe the sentence should be divided in two parts

8) page 4 line 28: unnecessary brackets

9) page 5 line 8-10: data is a plural word, "were", "do"

10) page 5 lines 11-12: if the quality of the measurements is lower then it's half the accuracy (not double), the error doubles, the accuracy halves; numbers should go before the "for"

11) page 5 line 21: misspelling "turned"

12) page 6 line 10: sentence in brackets should actually be better illustrated and high-lightened since it's an important piece of information for the whole paper

13) page 6 line 22: unnecessary brackets

14) page 8 line 9: "by" not "from"

15) page 8 lines 9-12: unnecessary brackets; The vertical structure of the eddy is unclear: What is the depth of the euphotic zone and how does it compare with the depth of the eddy core? What is the depth of the mixed layer? Is primary production only taking place in the shallow mixed layer as it may be hypothesized from the chlorophyll plot, or is primary production also happening in the core? Is the core still in the euphotic zone? These points should be very well clarified also in the "Results" sections

16) page 8 line 13: misspelling "resembles"; I find it not so proper to say that this sporadic upwelling resembles coastal upwelling, it's probably Ekman pumping, which does not require a coastal boundary to happen

17) page 8, lines 17-31: this description should in part be moved to an eddy description section and in part be included in the 2.2 Reference Data Sets section

18) page 8 lines 23-25: sentence is hardly understandable, "but" is incorrectly used

19) page 8 line 29: "en route"?

20) page 9 lines 4-6: "remineralization", not "mineralization"; words before acronyms should be capitalized; "age" doesn't need quotation marks; very convoluted sentence, could be split in two parts

21) page 10 line 9: there should be a comma (not a dot) before "as"

22) page 10 line 24: misspelling "resemble"

23) page 10 line 25: "predominating" not the right word, maybe "dominant"

24) page 10 lines 26-27: not sure if this sentence is needed. Either the paper includes a full description of the basin and relative water masses (eg, in the introduction) or it seems out of place; also: what is the typical TS signature of SACW that is also found in the core of the eddy? A reader my not be familiar with this water mass

25) page 11 line 8: I don't understand "vertical contrast", does it mean "gradient"?

26) page 11 line 11: "underway"?

27) page 11 line 22: "minimal" not an adverb

28) page 11 lines 25-26: "inside the ACME" seems to refer to the whole eddy, are these lines referring to remineralization that happens in the low-oxygen core?

29) page 12 line 25: clearly missing reference (!)

30) page 13 line 5: paragraph on DIC should probably end here, not at line 3

31) page 13 line 6-7: not quantitative, not clear, terms as "minor change" and "small

but significant" should be defined

32) page 13 lines 17-18: "data not shown" used for drawing conclusion does not strengthen the paper, given the limited number of plots and their simplicity maybe some of the data could also be shown; same for the next "data not shown" in the paper

33) page 13 lines 20-32: all the detected small particles are assumed to be POM; this assumption is not explicitly stated even though it is at the base of the conclusions, I suggest to state and justify the assumption for this region. Are dust-deposition-derived particles irrelevant in this region/season?

34) page 13 line 23: convoluted sentence

35) page 13 line 24 and 27: "significantly exceeds" and "much higher" not very quantitative

36) page 13 lines 30-31: "according to Hauss et al (2015)", is this the same eddy?

37) page 17, lines 22-32: In the first paragraph of the conclusions the authors are very generic about their findings regarding the eddy bgc composition discussed in the sections from 3.1 to 3.4. Some of the results are not recalled (eg, particle, POM and DOM distribution). Since many interesting findings are discussed in the paper, I strongly suggest strengthening this part of the conclusion.

38) Figure 2: colorbar and legend seem to contradict each other: if orange points refer to ACME (M105), what do the blue points refer to? This choice of colormap is unhelpful,

the color is mostly constant.

---

## Author Comment (AC1) · 20 May 2016

Dear Reviewer,

First of all we would like to thank you for the time and effort you spent on reviewing our manuscript. We very much appreciate your comments that clearly have identified parts of the paper that needed more attention. We tried to address all the questions and comments you raised and are convinced that the manuscript will be improved significantly.

In the following we sorted all comments/questions (**RC:**) by numbering these (according to your numbering) and providing for each an answer (**AC:**).

Best regards,
B. Fiedler & Coauthors

**1. Reviewer #1:**

Review of Fiedler et al., Oxygen Utilization and Downward Carbon Flux in an Oxygen-Depleted Eddy inb the Eastern Tropical North Atlantic.

bg-2016-23

**General comments:**

**Reviewer Comment (RC)1_:** *Boundary definition: The authors are rather vague about the definition of eddy boundaries (the lateral boundaries, but in particular the lower boundary of the eddy core), as well as the processes leading to exchanges, or lack thereof, across those boundaries. I would suggest the authors provide more details on eddy boundaries, the depth of the mixed layer, and the depth of euphotic zone, as well as provide a stronger case for rationalizing why eddy waters don't mix with the surrounding ocean waters. For example, is the depth of the euphotic zone (Dez) and mixed layer depth in the eddy core equal to the outside water? Or is Dez different due to higher light attenuation by particles?*

**Author Comment (AC)1_:** We agree that details about the physical boundaries of this eddy are not well described in this manuscript. However, we left this out on purpose as Karstensen et al. (2016, this special issue) are presenting an elaborate physical analysis of boundaries for this particular eddy. But we have to admit that this link was not made clear enough. Therefore we will add a sentence to section 3.1:

"Isolation of this eddy was found to be caused by high eddy rotation speed and stratification and their joint impact on the propagation of internal waves (Karstensen et al., 2016, this special issue)."

We also edited two sentences in section 3.1 in order to explicitly mention the mixed layer depth in and outside the eddy.

"The upper bound of the eddy core is the mixed layer base at a depth of 70 m which has the same magnitude as the mixed layer outside the eddy (Karstensen et al., 2016, this special issue). A very sharp gradient exists between 70 – 77 m depth which amounts to 0.73 in salinity, 3.98°C in temperature and 165.8 µmol kg$^{-1}$ in dissolved oxygen."

Unfortunately, light/PAR measurements failed during the surveys due to sensor problems. Thus, we can't give reliable information about the euphotic zone for this particular eddy. We removed speculative connections between the eddy core and the euphotic zone from the abstract and the conclusions.

**RC2_:** *Episodic events: Throughout the paper a steady state biogeochemical system is implied (or at least a slowly evolving biogeochemical state). Yet the physical processes that allow for these balances are episodic and submesoscale (e.g. evidence for the re-supply of nutrients into the upper layer is lacking, and yet required for the equilibrium biogeochemical state in the mixed layer). Is this vertical nutrient flux driven by interaction of eddies with the overlying wind field, Ekman pumping, or internal waves displacing isopycnals and mixing? How frequent are these episodic events?*

**AC2_:** This is indeed an important point. Unfortunately, these processes are extremely difficult to observe, even with the tools we applied during this study (autonomous glider, see Karstensen et al., 2016, this special issue). We disagree that evidence for re-supply of nutrients into the upper layer is lacking as we clearly found elevated nutrient concentrations in the surface during one of the two ship surveys (section 3.2, 2$^{nd}$ paragraph). Since methodological biases of these samples can be ruled out, this observation can only be explained by an upward vertical flux. Observations for chl-a also indicate elevated levels of phytoplankton towards the upper part of the mixed layer, likely being facilitated by upwelling of subsurface waters. Finally, Karstensen et al. (2016, this special issue) derived a physical concept which provides a mechanism for upwelling occurring at the rim of the eddy, followed by horizontal distribution. The process is mainly driven by

downward propagation of internal near inertial waves. Since details about this mechanism are described in Karstensen et al. (2016), we added a sentence in section 3.2 (2ⁿᵈ paragraph) in order to provide a link to that paper:

"As such, this finding is interpreted as being a signature of a vertical flux event. The physical mechanism is described in more detail in Karstensen et al. (2016, this special issue)."

Further, we cannot derive information about how intermittent/sporadic the upwelling is and we will rephrase that to a more general statement on "upwelling processes". The analysis of an oxygen float (Karstensen et al., 2015) showed that the respiration derived from 5 day oxygen profiles over a period of several month showed a surprisingly constant decrease which in turn suggests that particle sinking is also constant and probably also the upwelling to ensure a constant bloom. However, the finite duration of a bloom also applies a "running mean" to any intermittence of the upwelling. This is definitely a process that requires further attention.

**RC3_:** *Downward carbon flux - POC export: This seem to be the weakest part of the paper. The authors need to provide more details on physical and biological assumptions in the simple downward flux model, and whether its assumptions are valid in the eddy core. Is a steady state balance implied for model? A constant diffusivity? What about small non sinking POC export by eddy flow field subduction of surface waters with high POC concentrations? (See Mahadevan et al., 2015, Science). Consequently, the carbon flux model and Figure 8 may not make a meaningful contribution to the study.*

**AC3_:** The reviewer addresses the right issues if the applied model would have the intention to fully explain the total vertical carbon fluxes. We have to admit that the way we introduced the downward POC flux calculations might be a bit misleading and makes the reader to expect a more comprehensive model that also accounts for physical transportation processes such as diffusion or subduction. Our main intention, however, was to look whether observed carbon remineralization of sinking particles inside the core follows a classical Martin curve and how this amounts to an export of POC out of the euphotic zone over the lifetime of this eddy. In order to assess the magnitude of observed carbon remineralization compared to other ocean environments (e.g., open ocean ETNA, coastal upwelling, North Atlantic, eddies) we decided to translate our results to a mean daily POC flux. Our results suggest that the Martin curve fits well the carbon remineralization rates and thus sinking particles are likely to be the major driver for our observations. The 3-fold enhanced carbon export also matches very well with the independently determined production rates which were also enhanced by a factor of 3 (Löscher et al., 2015).

Regarding the subduction of high POC waters as described in Omand et al. (2015) we think that this mechanism would not affect our POC calculations for two reasons: 1) Described subduction in Omand et al. occurs rather at the perimeter of the eddy and 2) the special physical conditions that characterizes an ACME would not allow for an intrusion of subducted (high POC) waters into the ACME core.

Finally, given that the core of the eddy is a transient system (oxygen consumption without reventilation) and only the limited number of observations in space and time we won't be able to come up with a more detailed approach.

Thus, we use a simplistic approach to observed change in biogeochemical water properties in order to give quantitative estimates of the POC flux and carbon export. For more sophisticated export flux models we lack information but this does not mean that carbon export can be fully explained by a Martin-type function. In general, however, the models fit our data well and is thus only used to extract some quantitative information for further comparison. We therefore think that this section still make a meaningful contribution to the manuscript. In order to clarify this topic we edited the beginning of section 2.5 as follows: "In order to estimate the amount of carbon exported from the euphotic zone as sinking POM we used CRRs to derive the shape of the vertical export flux curve for particulate organic carbon (POC). This approach assumes the

absence of major physical transport processes between the mixed layer and the ACME core beneath except for sinking particles of POM which is generally being described by the established Martin Curve (Martin et al., 1987a):"

**RC4_:** *Significance (and a Budget): What is significance of 1 to 2 ACMEs generated every year that propagate into the open ETNA waters for biogeochemistry and ocean acidification of the region, or even the ocean basin? Is this a phenomenon that might be expect to occur elsewhere (does the study have much broader implications? To help address this it might be worth carrying out at a budget/balance exercise to quantify, for example, whether the supply of nutrients exceeds export of nutrients, which leads to than increased productivity.*

**AC4_:** To estimate the significance of these eddies was also one of our main objectives of the "Eddy Hunt" project the special issue is concerned with. Besides a local impact, that is important for a process understanding, the larger scale impact, at least for the eastern tropical Atlantic is of interest. The occurrence of the dead-zone eddies was analyzed in a study submitted in parallel to ours(Schütte et al., 2016, this special issue). The authors analyzed satellite (SLA, SST) and in-situ oxygen and T/S profile data for the eastern tropical North Atlantic. They estimated that about 1 to 2 low oxygen eddies disperse in the region every year but because of the anomalous low oxygen they found an at least 6% contribution of these low-O2 ACMEs in the maintenance of the shallow oxygen minimum zone (centered at about 70m depth and about 250m above the core OMZ). The conclude that their estimate is a conservative one since the detection of ACMEs from satellite data is challenging (because of a weak SLA signature of ACMEs) and the actual number of ACMEs is likely to be higher. We will add a sentence at the end of the 2$^{nd}$ paragraph of the conclusion as follows:

"As revealed by Schütte et al. (2016) these ACMEs appear to play a small but significant role in maintaining the shallow OMZ in the ETNA."

Probably related to the improvement of multidisciplinary autonomous and high resolution satellite-borne observing techniques, much attention has been devoted to investigating local processes in and large scale impact of ocean mesoscale eddies. Many studies have recently been published. What is specific for the ETNA region is the cold/fresh core of the eddies, an imprint from the coastal current and definitely different from the Pacific, where the coastal current carries warm/saline water. This could matter for the temporal evolution of the stratification and as such for the isolation (see Karstensen et al. 2016). For the purpose of this paper we limited our regional examples of other studies to the North Atlantic (McGillicuddy et al., 2007) and the South Pacific (Altabet et al., 2012).

**Specific comments:**

**Abstract:**
**RC1**: Page 1, Line 13: Define the extreme low oxygen environment.
**AC1**: Sentence edited: "The occurrence of mesoscale eddies that develop suboxic environments at shallow depth (about 40 to 100 m) has recently been reported for the eastern tropical North Atlantic (ETNA)."

**RC2:** Page 1, Line 22: Define the lower boundary of the euphotic zone.
**AC2**: Changes mentioned under AC1_, see above.

**RC3:** Page 1, Line 27: Define the lower boundary of the surface mixed layer. Is this shallower than the euphotic zone?

**AC3**: Information will be added as: "Vertical distributions of particulate and dissolved organic matter (POM, DOM) generally show elevated concentrations in the surface mixed layer (0 – 70 m), but particularly DOM also accumulates beneath the oxygen minimum."
Direct observations of light attenuation are missing, see also AC1_.

**RC4:** Page 2, Line 1: an enhancement of apparent oxygen utilization rates: : :at what depth?
**AC4**: We will modify two sentences of the abstract to define the depth of the eddy core as follows:
"At the time of the survey the eddy core showed lowest oxygen concentrations of less than 5 $\mu$mol kg$^{-1}$ and a pH of approx. 7.6 at a depth of approx. 100 m."
"Considering reference data from the upwelling region where these eddies are formed, we determined the oxygen consumption through remineralization of organic matter and found an enhancement of apparent oxygen utilization rates (aOUR, 0.26 $\mu$mol kg$^{-1}$ d$^{-1}$) inside the core by almost one order of magnitude when compared with typical values for the open North Atlantic."

**RC5:** What is the significance of your findings for the biogeochemistry of ETNA?
**AC5**: As this study does not directly determine significance in a quantitative way we cannot define this in the abstract. However, we will edit the last sentence of the abstract as follows: "The observations support the view that the oxygen depleted eddies can be viewed as isolated, westwards propagating upwelling systems of their own and thereby represent re-occurring alien biogeochemical environments in the ETNA."

**Introduction:**
**RC6:** Page 2, Paragraph 2: A figure showing ETNA, OMZ, EBUS, CVFZ would be helpful. Maybe add on to Figure 1.
**AC6**: We appreciate this constructive comment from the reviewer and will add some of the acronyms to Figure 1 (both main and inlet panels).

**RC7:** Page 3, Line 9: Define anticyclone mode-water eddies.
**AC7**: We will follow this suggestion and add one sentence after the introduction of ACMEs: "The latter ones are characterized by a water lens of mode which is being formed by up- and downward-bent isopycnals towards the eddy center."

**RC8:** What is the main objective of this paper?
**AC8**: We will add one sentence to the last paragraph of the introduction: "Here, we present the first biogeochemical insights into low-oxygen ACMEs in the ETNA based on direct in situ sampling during two coordinated ship-based surveys. The main objective of this study is to reveal and quantify biogeochemical processes occurring inside a low-oxygen ACME in the ETNA."

**Methods:**
**RC9:** Page 5, Line 8: At what depth were water samples collected?
**AC9**: Information will be added: "Water samples in the upper 500 m were collected with a rosette water sampling system…"

**RC10:** Page 6, Line 26: At what depth were the water samples for DO collected?
**AC10**: Niskin bottles during both cruises were closed following a certain depth grid. Only depths of Niskin bottles close to the eddy core/oxygen minimum were slightly adjusted in order to adequately resolve this part of the water column. All samples (for various parameters) were collected for each depth. Since sampling depths for each parameter can be also deduced from figures 4 – 6 we decided to not explicitly mention this information in the text.

**RC11:** Page 7, Line 10: same as above, define depth of DIC and TA samples.

**AC11**: see above (AC10).

**RC12:** Page 7, Lines 21-24, using which software?

**AC12**: We will add this information to the text (and add the respective reference) as follows: "Results from DIC and TA analysis were used to compute the remaining parameters of the marine carbonate system (pH, pCO2 and ΩAr) using a MATLAB version of the CO2SYS software (Van Heuven et al., 2011). Calculations were based on carbonic acid dissociation constants after Mehrbach et al. (1973) as refitted by Dickson and Millero (1987)."

**RC13:** Page 9, Line 4: Define apparent oxygen utilization rate.

**AC13**: We will modify the respective section as follows: "Changes of oxygen and carbon due to remineralization of organic matter are being expressed as the apparent oxygen utilization rate (aOUR) and the carbon mineralization rate (CRR). In order to determine these rates not only the anomaly but also the age of the eddy, the time between formation on the shelf and the time the eddy surveys took place, needs to be known."

**Results and Discussion:**
**RC14:** Page 10, Lines 15-20: This text is repetitious, and can be omitted.
**AC14**: We agree and will remove this part from the manuscript.

**RC15:** Page 10, Lines 22-24: Specify depth.
**AC15**: Since the sentence describes the characteristics of ACMEs in the ETNA in general we can't specify a certain depth. However, we will add the word "subsurface" to this sentence: "The Temperature-salinity (TS) characteristics of the subsurface core of ACMEs in the open ETNA (Schütte et al., in prep. for this issue; Karstensen et al., 2015) were found to be nearly unchanged, compared to coastal regions."

**RC16:** Page 11, Line 20: (100m)- not clear if is this at 100m, or from 0 to100m?
**AC16**: We will slightly rephrase this sentence to avoid potential confusion: "In comparison to the reference profile from the Mauritanian Shelf we find a maximum oxygen decrease in the eddy core at a depth of 100 m of about 57.0 µmol kg-1 to suboxic levels (<5 µmol kg-1; Figure 3)."

**RC17:** Page 11, Lines 16-18 (sentence 2): move to after describing the results.
**AC17**: Thank you for this remark. We decided to fully delete this sentence as it is redundant with the second last sentence of this paragraph.

**RC18:** Page 11, Line 27: Why elevated nitrate, nitrite and phosphate but not silicate? Also, did you look into nitrate:phosphate ratio as evidence of denitrifying bacteria?
**AC18**: We interpret the depletion of silicate as a consequence of high abundance of diatoms in the surface mixed layer. From sediment trap data at CVOO (Fischer et al., 2015, this special issue) we know that diatoms were the dominant species during the passage of an ACME in 2010 at CVOO. Koeve (2004) also reported on high nitrate:silicate ratios in the North Atlantic and explains this with a different uptake ratios than the nitrate:silicate ratio of upwelled waters. We extended the discussion of surface nutrients as follows: "In contrast, silicate concentration remained low which could be explained by an enhanced abundance of diatoms in the mixed layer. Further, Fischer et al. (2016) reported on high opal concentrations in sediment traps at CVOO which were associated with the passage of a former ACME passing the observatory. High N:Si uptake ratios, also reported for the North Atlantic (Koeve, 2004), could explain observed nutrient concentrations."
We also looked into nitrate:phosphate ratios but couldn't find deviations pointing towards denitrification. However, genetic analysis of the microbial community clearly revealed active

denitrification in the core of the eddy. N:P ratios as well as the discussion on denitrification of this particular eddy are already published (Löscher et al., 2015).

**RC19:** Page 12, Lines 1-5: How often do these sporadic events occur? Are they wind induced, or due to passage of internal waves?
**AC19**: Please see above (AC2_).

**RC20:** Page 12, Lines 7-9: This sentence is a repetition from Methods and can be omitted.
**AC20**: We fully agree and will omit this sentence.

**RC21:** Page 12, Line 26: "phytoplankton communities are exposed to these acidified conditions". How often? (the pH minimum is located just below the euphotic zone).
**AC21**: The reviewer is right in asking about abundance of phytoplankton in the core at this depth (beneath or close to the euphotic depth). However, here we wanted to give a general statement about acidified cores of ACMEs. Indeed, the particular ACME surveyed during this study has its core relatively deep. Other ACMEs may have more shallow cores which are located within the euphotic zone. Karstensen et al. (2015), for instance, observed an ACME in the same region in which the core reached up to even 40 m water depth which is very likely to be within the euphotic zone. In order to keep the respective sentence more general we will slightly edit it as follows: "Absolute values of pH inside the eddy exceed these predictions and plankton communities inside shallow low-oxygen cores of ACMEs may get exposed to these acidified conditions."

**RC22:** Page 13, Lines 25-29: is this consistent with your Chl-a observations?
**AC22**: This finding is consistent with discrete samples for chl-a which are presented in Löscher et al. (2015). Chl-a concentrations as illustrated in Figure 3 (most right panel) were derived from different fluorescence sensors mounted at the rosette water samplers during different cruises. Computation of chl-a concentrations were based on factory calibrations for each individual sensor. The fact that calibration of fluorescence data for the determination of chl-a concentration is not well developed and daylight quenching of fluorescence at the surface (Xing et al., 2012) biases this data as well, we doubt that absolute concentrations derived from fluorescence sensors are robust for a quantitative interpretation. We rather see these sensor measurements as a qualitative proxy that describes rather the vertical distribution of phytoplankton in the water column.
We will add two sentences to the methods section as follows: "Additional sensors such as an oxygen sensor (SBE43, Seabird Electronics) and a two channel fluorometer (FLNTURT, WETLabs) were attached to the CTD. Note that factory-calibrated fluorometer data in this study can be only used as a qualitative proxy for phytoplankton distribution in the water column due to a lack of elaborate sensor calibrations."

**Conclusions:**
**RC23:** Page 17, Line 30: "intense increase": : :specify where.
**AC23**: We will add this information to the sentence as suggested: "Likewise, nutrient concentrations as well as pCO2 levels showed a large increase within the eddy core which created significant anomalies when compared to ambient open-ocean ETNA conditions."

**RC24:** Page 18, Line 9: Is the 3-fold increase in primary productivity consistent with your observed Chl-a?
**AC24**: Please refer to AC22.

**RC25:** Page 18, Lines 20-25: I would replace this text with a sentence on the significance of the 1 to 2 ACMEs generated every year that propagate into the open ETNA waters for biogeochemistry and ocean acidification of the area.

**AC25**: Thank you for this suggestion. However, please see AC4_ for details – as we decided to not replace this text with findings of Schütte et al. (2016), as at this part of the conclusions we explicitly want to look forward and give an outlook about open questions which need to be addressed by future studies. Instead, we will add a sentence to the end of the second paragraph of the conclusions as follows: "As revealed by Schütte et al. (2016) these ACMEs appear to play a small but significant role in maintaining the shallow OMZ in the ETNA."

**Tables:**

**RC26:** Table 1: Add: negative values correspond to: : : Are these average anomalies over some depth range?

**AC26**: We will edit the caption as suggested: "Overview of detected concentration anomalies ($\Delta$total) within the ACME core ($\sigma\theta$ =26.35 kg m-3 - 1000) during the two surveys referenced against prevailing conditions at the shelf. Rate estimates are based on the lifetime of the ACME derived from satellite sea level anomaly data (ISL: 163 days, M105: 173 days). Values for the average shelf profile are given in order to illustrates local variability at the corresponding isopycnal (=26.35 kg m-3 - 1000). Negative values correspond to a decrease of the respective parameter over the lifetime of the ACME."

**Figures:**

**RC27:** Figure 2: SACW missing?

**AC27**: Both water masses are represented by gray dashed lines. We find the gray scale sufficient to get noticed both in its digital and printed form. However, we will double this once it comes to the technical processing of this manuscript.

**RC28:** Figure 8: This figure and the associated carbon flux model (eqn 2) does not make a significant contribution to the paper as it stands. See general comment #3 above.

**AC28**: Please refer to AC3_.

**References:**

Altabet, M. A., Ryabenko, E., Stramma, L., Wallace, D. W. R., Frank, M., Grasse, P. and Lavik, G.: An eddy-stimulated hotspot for fixed nitrogen-loss from the Peru oxygen minimum zone, Biogeosciences, 9(12), 4897–4908, doi:10.5194/bg-9-4897-2012, 2012.

Fischer, G., Karstensen, J., Romero, O., Baumann, K.-H., Donner, B., Hefter, J., Mollenhauer, G., Iversen, M., Fiedler, B., Monteiro, I. and Körtzinger, A.: Bathypelagic particle flux signatures from a suboxic eddy in the oligotrophic tropical North Atlantic: production, sedimentation and preservation, Biogeosciences Discuss., 12(21), 18253–18313, doi:10.5194/bgd-12-18253-2015, 2015.

Karstensen, J., Fiedler, B., Schütte, F., Brandt, P., Körtzinger, A., Fischer, G., Zantopp, R., Hahn, J., Visbeck, M. and Wallace, D.: Open ocean dead zones in the tropical North Atlantic Ocean, Biogeosciences, 12(8), 2597–2605, doi:10.5194/bg-12-2597-2015, 2015.

Karstensen, J., Schütte, F., Pietri, A., Krahmann, G., Fiedler, B., Grundle, D., Hauss, H., Körtzinger, A., Löscher, C. R., Testor, P., Vieira, N. and Visbeck, M.: Upwelling and isolation in oxygen-depleted anticyclonic modewater eddies and implications for nitrate cycling, Biogeosciences Discuss., 1–25, doi:10.5194/bg-2016-34, 2016.

Koeve, W.: Spring bloom carbon to nitrogen ratio of net community production in the temperate N. Atlantic, Deep Sea Res. Part I Oceanogr. Res. Pap., 51(11), 1579–1600, doi:http://dx.doi.org/10.1016/j.dsr.2004.07.002, 2004.

Löscher, C. R., Fischer, M. A., Neulinger, S. C., Fiedler, B., Philippi, M., Schütte, F., Singh, A., Hauss, H., Karstensen, J., Körtzinger, A., Künzel, S. and Schmitz, R. A.: Hidden biosphere in an oxygen-deficient Atlantic open-ocean eddy: future implications of ocean deoxygenation on primary production in the eastern tropical North Atlantic, Biogeosciences, 12(24), 7467–7482, doi:10.5194/bg-12-7467-2015, 2015.

McGillicuddy, D. J., Anderson, L. A., Bates, N. R., Bibby, T., Buesseler, K. O., Carlson, C. A., Davis, C. S., Ewart, C., Falkowski, P. G., Goldthwait, S. A., Hansell, D. A., Jenkins, W. J., Johnson, R., Kosnyrev, V. K., Ledwell, J. R., Li, Q. P., Siegel, D. A. and Steinberg, D. K.: Eddy/Wind Interactions Stimulate Extraordinary Mid-Ocean Plankton Blooms, Science (80-. )., 316(5827), 1021–1026, doi:10.1126/science.1136256, 2007.

Omand, M. M., D'Asaro, E. A., Lee, C. M., Perry, M. J., Briggs, N., Cetinić, I. and Mahadevan, A.: Eddy-driven subduction exports particulate organic carbon from the spring bloom, Sci. , 348 (6231 ), 222–225, doi:10.1126/science.1260062, 2015.

Schütte, F., Karstensen, J., Krahmann, G., Hauss, H., Fiedler, B., Brandt, P., Visbeck, M. and Körtzinger, A.: Characterization of "dead-zone " eddies in the tropical Northeast Atlantic Ocean, Biogeosciences Discuss., (Special Issue), 2016.

Xing, X., Claustre, H., Blain, S., D'Ortenzio, F., Antoine, D., Ras, J. and Guinet, C.: Quenching correction for in vivo chlorophyll fluorescence acquired by autonomous platforms: A case study with instrumented elephant seals in the Kerguelen region (Southern Ocean), Limnol. Oceanogr. Methods, 10(7), 483–495, doi:10.4319/lom.2012.10.483, 2012.

---

## Author Comment (AC2) · 20 May 2016

Dear Reviewer,

First of all we would like to thank you for the time and effort you spent on reviewing our manuscript. We very much appreciate your comments that clearly have identified parts of the paper that needed more attention. We tried to address all the questions and comments you raised and are convinced that the manuscript will be improved significantly.

In the following we sorted all comments/questions (**RC:**) by numbering these (according to your numbering) and providing for each an answer (**AC:**).

Best regards,
B. Fiedler & Coauthors

**1. Reviewer #2:**

Review of Fiedler et al., Oxygen Utilization and Downward Carbon Flux in an Oxygen-Depleted Eddy inb the Eastern Tropical North Atlantic.
bg-2016-23

**General comments:**

**Reviewer Comment (RC)1_:** *Reference Data Set: an assumption that needs to be strengthened*
*The three Reference Data Sets used by the authors for the bgc fluxes calculations are from cruises that were conducted on the Mauritanian shelf around the second-half of July 2006, beginning of June 2010, and beginning of June 2014. The surveyed eddy is supposed to have formed on the Mauritanian shelf around June/July 2013. This can be inferred from the paper, but it's not explicitly stated. At page 8 lines 17-21 the authors explain how they reconstructed the region of origin of the eddy on the shelf on the base of statistical analysis of historical SLA, and how this region coincides with the location of the 3 Reference Data surveys. However, the eddy trajectory from SLA in "Figure 1" starts about two months later (Sept. 2013) at least 100km in the off-shore direction. The fact that the trajectory of this specific eddy was not retrieved on the shelf that may imply, for example, that the eddy boundaries of this eddy were not already well formed, therefore the eddy may have continued to trap water while leaving the shelf area of the Reference Data Sets, or may have been spun by a lateral filament. At page 11 lines 4-5 the authors underline the matching between the Temperature and Salinity of the eddy core when it was surveyed and the reference station measurements on the shelf. This supports the assumption of isolation of the eddy core from the shelf to the offshore waters. However, biogeochemical properties can be more variable than physical properties on both spatial and time scales, especially in active shelf regions. The authors write about the Reference Data Sets [page 8 lines 23-27] "in order to account for small scale variability [: : :] an average profile for each investigated parameter was created [: : :]. These mean profiles were assumed to represent typical initial conditions of ACMEs [: : :]". Given the complex dynamics of the flow around the shelf edge, the time and spatial variability of biogeochemical processes, the timescale of sporadic upwelling events; given the fact that the eddy trajectory from SLA was not retrieved in the shelf region, and the complexity of the eddy formation process:*

*1. Can the choice of these Reference Data Sets be better justified? Are there no available data for the region in which the trajectory was actually retrieved in Sept.2013?*
*2. How do mean profiles account for small scale variability?*
*3. Is it possible to exclude strong discontinuities (input of external water, sediment resuspension, interaction with other forming eddies, etc.) in the eddy evolution between the shelf region of the Reference Data Sets and beginning of the track in "Figure 1"?*
*4. Several times in the article the authors refer to sporadic upwelling events fueling the high surface productivity in the eddy. How is the hypothesis of "production being boosted in the surface of the eddy by upwelling events" compatible with the hypothesis of "complete isolation of the eddy core" along the whole eddy lifetime? What is the spatial distribution of these upwelling events in the eddy?*

**Author Comment (AC)1_:**
We appreciate this comment and related thoughts about proper reconstruction of initial conditions of the surveyed ACME. We also see the need to constrain initial conditions of the eddy as good as possible as this directly affects derived rates for biogeochemical parameters. According to RC17 we decided to slightly reorganize the presentation and discussion of SLA results. We will introduce the SLA-derived trajectory already under 2.2 (reference data sets) and will add some discussion on this under a more general eddy description section under 3 (results & discussion), as also proposed in RC2_.

We explicitly neglected SLA trajectory data closer to the shelf for two reasons: ACMEs only show a very minor sea level elevation of the eddy surface. This makes it very difficult to track these eddies. Furthermore, the high density of (short-lived) eddies and filaments close to the shore drastically impairs the reliability of the eddy tracking. Background noise impedes clear identification of individual eddies, in particular those with only minor signals (ACMEs). Schütte et al. (2016) performed an elaborate analysis of eddy statistics which clearly indicated this region as a release hotspot and states that this is related to a seasonal weakening of the coastal undercurrent along with coastal topographic features. Further, Thomsen et al. (2016) observed the initial formation of an ACME off Peru, which exactly took place at the shelf edge and thereby capturing hydrographic and biogeochemical conditions at this place. We cannot derive information about how intermittent/sporadic the upwelling is and we will rephrase that to a more general statement on "upwelling processes". The analysis of an oxygen float showed that the respiration derived from 5 day oxygen profiles over a period of several months showed a surprisingly constant decrease which in turn suggests that particle sinking is also constant and thus upwelling. However, the duration of the blooms apply a running mean to any intermittence of the upwelling.

Answers to the 4 specific issues:

1. We only have a very few surveys near the Mauritanian shelf edge available which fall into boreal summer months and which also conducted biogeochemical samplings in that region. Data are also available for regions further offshore from the same expeditions. However, as we are confident that the origin of the eddy is located closer to the shelf edge, we would significantly bias our calculations if you choose the more offshore area as the starting conditions.

2. This sentence was not phrased correctly in the discussion paper. We will rephrase is as follows: "In order to neglect small-scale variability of water column properties within this area, an average profile for each investigated parameter was created by averaging on isopycnals but mapped back to depth via the mean depth/density profile."

3. Even though we were not able to have in situ observations of this particular eddy close to its origin we are confident that once the eddy has been created no further exchange of waters masses between the inner and outer part took place. Usually, such eddies begin their lifetime with very stable conditions and slowly decay over their lifetime. Since we observed very stable conditions still after 6-7 months we don't think that major fluxes occurred in the early days of this eddy.

4. .       Indeed the apparent contrariety between isolation on the one side and upwelling on the other side is a very interesting observation. We do not have a final answer to it but in Karstensen et al. (2016, "Upwelling and isolation in oxygen-depleted anticyclonic mode water eddies and implications for nitrate cycling") we discuss a concept for the processes that interact on the submesoscale. In brief, the upwelling occurs at the rim of the eddy where the vertical shear in velocity is largest (enhanced by vertical propagating Near Inertial Internal Waves). The upwelling is thus expected to originate from shallow depth, say the upper 100m or so and should be rather constant. One part of the upwelled waters is "trapped" by eddy retention and as is accessible for productivity across the eddy. While we speculated in the past that the isolation is related to the eddy coherence further analysis reveals that the buoyancy frequency/stability maximum encompassing/defining the core is very efficient in isolating the core for mixing (e.g. shown in Sheen et al. (2015). Details are described in Karstensen et al. (2016). We will add a sentence in section 3.2 (2nd paragraph) in order to provide a link to that paper: "As such, this finding is interpreted as being a signature of a vertical flux event related to submesoscale processes and stratification which on the one side isolate the core and prevent oxygen supply while in parallel support vertical nutrient flux at the eddy rim (see Karstensen et al., 2016, this special issue, for further details)."

**RC2_:** *Description of the surveyed eddy*

*The authors do not provide a clear general description of the characteristics of the surveyed eddy, among which some basic details: date (month/year) and coordinates of the eddy when forming on the shelf; date and coordinates of the beginning of the track; date, coordinates, radius, shape and age of the eddy when surveyed. Some of these characteristics can be retrieved in different parts of the article explicitly or implicitly, but it is the work of the reader to collect them. I suggest presenting these characteristics in a dedicated paragraph where the eddy is introduced and described. As regard to "Figure 1": it may be helpful to add a timescale of the eddy trajectory and to draw the eddy contours in the region of the cruises, to understand where the measurements were located with respect to the eddy center and boundaries. Most of the observations described in the chapter "3 Results Discussion" would be much easier to understand if a nice description of the vertical physical structure of the surveyed eddy was given.*

**AC2_:** We appreciate this comment and will provide a dedicated section that describes the eddy characteristics. In this section we will provide the requested information and we will incorporate section 3.1 (Hydrography) as well. The new section will be entitled as "Eddy Characteristics" and will replace the current section 3.1 (Hydrography).

Regarding Figure 1 we will add a few dates for illustrating the timescale of the eddy propagation. However, we decided to not draw assumed eddy contours into the figure for the following reasons: 1) In reality the form factor of such an eddy is quite dynamic and a representing ellipse or circle would be a bit misleading, 2) the figure may become too busy and 3) this kind of illustration is already presented in two more papers as part of this special issue (Hauss et al., 2016; Löscher et al., 2015) and we want to avoid too many replicates. Finally, as our analysis mainly focuses on the comparison between the inner station, the CVOO reference station (far out of the eddy) and the shelf station (even further away) we don't see an urgent need to include the potential size and shape of the eddy to this figure.

**RC3_:** *Description of the dedicated eddy surveys*

*In paragraph "2.1 Eddy surveys" the number of samples that were collected during each cruise is not clear and some of the descriptions are confusing. As the 2 cruises are described as "first dedicated biogeochemical surveys" of the eddy the reader may expect to see some 2D biogeochemical sections and wonder what the spatial resolution of the samples is. However, it becomes clear later that the biogeochemical data analyzed in the article consist in only 2 bottle measurements per cruise, 1 per cruise referring to the eddy center. I suggest stating this clearly in the text. The CTD/UVP-only (no bgc) section M105 is introduced but never plotted or clearly discussed. At page 5 lines 19-24 the authors write that some stations supposed to be at a certain distance from the center of the eddy on the base of SLA "turned out probably more at the rim of the eddy than in the surrounding water representing typical background conditions". What is the reason for this conclusion? Can this be elaborated more in depth to justify this sentence?*

**AC3_:** We agree that we haven't clearly pointed out that the biogeochemical component of these surveys only comprises hydrocast stations in- and outside this eddy. Since we mostly focus on the eddy center stations (one during M105 and another one during ISL_00314) we will emphasize this in the text as follows: "During both cruises hydrographic and biogeochemical data were sampled in the same eddy (Figure 1) although extensive biogeochemical samplings were performed only during single hydrocast stations at the eddy center."

Since we don't show any data from the hydrographic section across the eddy (M105, see Hauss et al. 2016 or Löscher et al., 2015) we removed the short paragraph about this section and also removed the section from Fig1 in order to avoid confusion about this.

We will also add some evidence for the location of the outside station in relation to the eddy center and rim as follows: "Based on the SLA data the "outside stations" during ISL and M105 were located 43 and 54 kilometers away from the supposed eddy center, respectively. However,

ship-borne Acoustic Doppler Current Profiler data (ADCP; see Hauss et al., 2016) as well as SLA data (Löscher et al., 2015) suggest a radius of this eddy of approx. 50 - 55 km. This points out that these stations where more at the rim of the eddy than in the surrounding water representing typical background conditions."

**RC4_:** *Quantitative results*
*Some of the results presented in the sections from 3.1 to 3.4 are not well quantified. Data collected in the eddy center and data on the shelf or in the open Atlantic are often compared with not-well-defined or confusing terms. I strongly suggest giving to the descriptions a more quantitative flavor.*
**AC4_:** We will go through these sections carefully and remove these terms wherever possible (see also AC35).

**RC5_:** *Style: English, typos*
*The paper, apart for a few sections, is scattered with typos, misspellings and incorrect formulation of the English sentences. Sentences are often very convoluted and difficult to follow. The frequent use of bracketed subordinates makes the reading process even more complicated. I highly suggest a proof-reading of the paper for typos, grammar and syntax, as well as a simplification of the structure of the sentences and the limitation of the use of brackets to the very essential. Some errors are listed in the "Detailed comments" section.*
**AC5_:** We will follow the reviewer's recommendation by having the paper proofread by a second native speaker. We will remove bracketed subordinates wherever possible.

**Specific comments:**
**RC1**: Introduction: The section "Introduction" of the paper ends with a paragraph (from page 3 line 29, to page 4 line 10) that introduces the content of the article. However, it forgets to anticipate any section about consequences and conclusions of the present study. I suggest to strengthen this paragraph in this sense, anticipating to the reader the presence of relevant conclusions connected to the results.
**AC1:** We will edit this paragraph as follows: "Here, we present the first biogeochemical insights into low-oxygen ACMEs in the ETNA based on direct in situ sampling during two coordinated ship-based surveys. The main objective of this study is to reveal and quantify biogeochemical processes occurring inside a low-oxygen ACME in the ETNA. This publication is part of a series that describes biological, chemical and physical oceanographic processes and their interaction inside these eddies. In this publication we first present the vertical hydrographic structure of a surveyed ACME and discuss nutrients concentrations and the marine carbonate system. All data are put into regional context by comparing ACME conditions with 1) ambient background conditions represented by CVOO and 2) the biogeochemical setting in the proximal EBUS off the West African coast, where the eddy originated from. Derived estimates for transformation rates of various key parameters and for carbon export rates within the surveyed ACME highly exceed known values for the ETNA and also other open-ocean regions."

**RC2**: page 2 line 25: Eastern Tropical should be capitalized when defining acronym
**AC2:** We will change this as suggested.

**RC3**: page 3 lines 18-20: incorrect syntax, "that" should follow the name that it refers to (ACME)
**AC3:** Sentence will be corrected: "They found that about 2 to 3 ACMEs are generated each year at distinct regions in the EBUS and then propagate into the open ETNA waters."

**RC4**: page 3 line 27: commas out of place

**AC4:** Sentence will be corrected: "Consequences for carbon cycling such as production and export as well as the impact on the ETNA OMZ also remain unclear."

**RC5**: page 3 line 29: word "process" is redundant
**AC5:** The word "process" will be removed.

**RC6**: page 4 line 2: misspelling "describes"
**AC6:** Will be corrected.

**RC7**: page 4 lines 16-19: sentence beginning with brackets; confusing sentence, "in the ETNA" better after "in situ-data", maybe the sentence should be divided in two parts
**AC7:** We will correct and rephrase the sentence as follows: "Schütte et al. (2016) analyzed satellite and corresponding in-situ data in the ETNA and found that on average about 20% of all anticyclones (10% of all eddies) are ACMEs that exhibit a pronounced low oxygen core."

**RC8**: page 4 line 28: unnecessary brackets
**AC8:** Brackets will be removed.

**RC9**: page 5 line 8-10: data is a plural word, "were", "do"
**AC9:** Will be corrected as suggested.

**RC10**: page 5 lines 11-12: if the quality of the measurements is lower then it's half the accuracy (not double), the error doubles, the accuracy halves; numbers should go before the "for"
**AC10:** Will be corrected as suggested.

**RC11**: page 5 line 21: misspelling "turned"
**AC11:** Section will be rephrased according to AC3_.

**RC12**: page 6 line 10: sentence in brackets should actually be better illustrated and high-lightened since it's an important piece of information for the whole paper
**AC12:** We will remove this sentence and place this information at the beginning of this section: "Analysis of SLA data of the surveyed eddy revealed that it was generated in August/September 2013 close to the Mauritanian shelf (Figure 1)."

**RC13**: page 6 line 22: unnecessary brackets
**AC13:** Sentence will be split and rephrased as follows: "The observatory includes a ship-based sampling and a mooring program (Fischer et al., 2015; Karstensen et al., 2015). At the time of the ISL sampling CVOO was located about 167 kilometers south of the eddy survey location in an open-ocean setting."

**RC14**: page 8 line 9: "by" not "from"
**AC14:** Will be corrected as suggested.

**RC15**: page 8 lines 9-12: unnecessary brackets; the vertical structure of the eddy is unclear: What is the depth of the euphotic zone and how does it compare with the depth of the eddy core? What is the depth of the mixed layer? Is primary production only taking place in the shallow mixed layer as it may be hypothesized from the chlorophyll plot, or is primary production also happening in the core? Is the core still in the euphotic zone? These points should be very well clarified also in the "Results" sections
**AC15:** Brackets will be removed.

Regarding the vertical structure of the eddy we will provide this information in the "Eddy description" section following your suggestion made in RC2_. We also edited two sentences in section 3.1 (which will be incorporated into the new section) in order to explicitly mention the mixed layer depth in and outside the eddy.

"The upper bound of the eddy core is the mixed layer base at a depth of 70 m which has the same magnitude as the mixed layer outside the eddy (Karstensen et al., 2016, this special issue). A very sharp gradient exists between 70 – 77 m depth which amounts to 0.73 in salinity, 3.98°C in temperature and 165.8 µmol $kg^{-1}$ in dissolved oxygen."

Unfortunately, light/PAR measurements failed during the surveys due to sensor problems. Thus, we can't give reliable information about the euphotic zone for this particular eddy. We removed speculative connections between the eddy core and the euphotic zone from the abstract and the conclusions.

Results for primary production rates are presented in Löscher et al., 2015 (Figure 7). Rates were found to be in accordance with discrete samples for chl-a. Unfortunately, no rates were determined for the depth of the eddy core.

**RC16**: page 8 line 13: misspelling "resembles"; I find it not so proper to say that this sporadic upwelling resembles coastal upwelling, it's probably Ekman pumping, which does not require a coastal boundary to happen

**AC16:** It is true that coastal upwelling is maybe mostly an Ekman pumping problem, while for the eddies different upwelling models have been proposed. However, the effect of eddy-induced upwelling on the biogeochemistry is comparable to coastal upwelling regions (upward nutrient flux, enhanced surface productivity, etc.).

**RC17**: page 8, lines 17-31: this description should in part be moved to an eddy description section and in part be included in the 2.2 Reference Data Sets section

**AC17:** We appreciate this comment and will move most of this paragraph into section 2.2

**RC18**: page 8 lines 23-25: sentence is hardly understandable, "but" is incorrectly used

**AC18:** We will split and rephrase this sentence as follows: "In order to neglect small-scale variability of water column properties within this area, an average profile for each investigated parameter was created. This was done by averaging parameters along isopycnal surfaces and then mapping back these values to the mean depth of each isopycnal surface."

**RC19**: page 8 line 29: "en route"?

**AC19:** Sentence will be reworded as follows: "This reference data from the shelf was then used to determine the changes in biogeochemical parameters that occurred on the way from the formation to the survey area northwest of Cape Verde."

**RC20**: page 9 lines 4-6: "remineralization", not "mineralization"; words before acronyms should be capitalized; "age" doesn't need quotation marks; very convoluted sentence, could be split in two parts

**AC20:** Sentence will be split and corrected as suggested: "Changes of oxygen and carbon due to remineralization of organic matter are being expressed as the Apparent Oxygen Utilization Rate (aOUR) and the Carbon Remineralization Rate (CRR). In order to determine these rates not only the anomaly but also the age of the eddy, the time between formation on the shelf and the time the eddy surveys took place, needs to be known."

**RC21**: page 10 line 9: there should be a comma (not a dot) before "as"

**AC21:** Sentence will be changed as suggested.

**RC22**: page 10 line 24: misspelling "resemble"
**AC22:** Correction will be done.

**RC23**: page 10 line 25: "predominating" not the right word, maybe "dominant"
**AC23:** Word will be changed to "dominant".

**RC24**: page 10 lines 26-27: not sure if this sentence is needed. Either the paper includes a full description of the basin and relative water masses (eg, in the introduction) or it seems out of place; also: what is the typical TS signature of SACW that is also found in the core of the eddy? A reader my not be familiar with this water mass
**AC24:** We decided to mention NACW at this part of the paper as it directly emphasizes the observed anomaly. We will incorporate this sentence into the sentence prior to it: "They resemble South Atlantic Central Water (SACW), the dominating upper layer water mass in the Mauritanian Upwelling region, whereas the region around CVOO is actually dominated by high salinity North Atlantic Central Waters (NACW; Pastor et al., 2008)."

**RC25**: page 11 line 8: I don't understand "vertical contrast", does it mean "gradient"?
**AC25:** Term will be replaced by "gradient".

**RC26**: page 11 line 11: "underway"?
**AC26:** "underway" will be replaced by "Shipborne Sea Surface Temperature (SST)"

**RC27**: page 11 line 22: "minimal" not an adverb
**AC27:** Sentence will be changed as follows: "We expect the oxygen decrease from continuous respiration of the organic material that sinks out of the euphotic zone into an environment that is at most only slightly affected by lateral ventilation of the eddy waters."

**RC28**: page 11 lines 25-26: "inside the ACME" seems to refer to the whole eddy, are these lines referring to remineralization that happens in the low-oxygen core?
**AC28:** We will now write "ACME core" in order to avoid confusion.

**RC29**: page 12 line 25: clearly missing reference (!)
**AC29:** We will add Schulz et al. (2013) who describe the range of pH values used for their mesocosm study.

**RC30**: page 13 line 5: paragraph on DIC should probably end here, not at line 3
**AC30:** Will be changed as suggested.

**RC31**: page 13 line 6-7: not quantitative, not clear, terms as "minor change" and "small but significant" should be defined
**AC31:** We will add the maximum value of change in TA beneath the eddy core as follows: "Here, only a minor small change of up to 17 µmol kg-1 in TA inside the eddy core is found." We will further rephrase the following sentence as well: "This was expected as respiration processes may have a positive or negative effect on TA depending on the form of reactive nitrogen being released (Wolf-Gladrow et al., 2007)."

**RC32**: page 13 lines 17-18: "data not shown" used for drawing conclusion does not strengthen the paper, given the limited number of plots and their simplicity maybe some of the data could also be shown; same for the next "data not shown" in the paper
**AC32:** We decided to rather remove this sentence as the differences in correlations are very weak due to the very limited number of TA samples. Correlations pointing towards this direction but

are clearly not robust. Regarding the following "data not shown" we decided to keep this statement as it is. It relates to POC samples collected during the remaining part of the M105 expedition south of Cape Verde. If we would include all this POC data into the subpanel it would make this plot too busy.

**RC33**: page 13 lines 20-32: all the detected small particles are assumed to be POM; this assumption is not explicitly stated even though it is at the base of the conclusions, I suggest to state and justify the assumption for this region. Are dust-deposition-derived particles irrelevant in this region/season?

**AC33:** The UVP is an optical instrument, thus we do not know the composition of the respective size classes of particles. However, although dust deposition is certainly an important factor in this region (also for ballasting of particles containing organic compounds), it seems unlikely that the marked increase in particles observed within the eddy is linked to a local dust event rather than the eddy itself. Particles larger than approximately 500 µm equivalent spherical diameter (for which the UVP stores the image information) mostly resemble "marine snow"-type aggregates (compare section across eddy and example images in Hauss et al. 2016, Fig 4a). While they may contain lithogenic material to some extent, it seems reasonable that they contribute to the POM (which was also measured independently by elemental analysis in discrete bottle samples) and provide the basis for water column respiration and carbon export flux (see also Fischer et al. 2015). We will edit the first sentence of this section as follows: "We used data from the UVP to illustrate vertical distribution of small particles (60 – 530 µm) in the water column which we assume to primarily consist of POM but may also contain lithogenic material (Fischer et al., 2015)."

**RC34**: page 13 line 23: convoluted sentence

**AC34:** We suppose that actually the sentence in lines 21-23 was meant. Thus, we rephrased this sentence as follows: "During both surveys, particle abundances show a peak within the shallow OMZ slightly below the oxygen minimum (Figure 6)."

**RC35**: page 13 line 24 and 27: "significantly exceeds" and "much higher" not very quantitative

**AC35:** We will change this as follows: "This points at accumulated particles fueling microbial respiration in the core of the eddy. Furthermore, surface concentrations of particles exceed open-ocean conditions as found at CVOO by a factor of 2 to 3. This is in line with Löscher et al. (2015) who described a threefold higher primary production for surface waters inside the eddy as compared to the outside. In the Mauritanian shelf area particle concentrations are high throughout the water column (Figure 6)."

**RC36**: page 13 lines 30-31: "according to Hauss et al (2015)", is this the same eddy?
**AC36:** Yes, it is.

**RC37**: page 17, lines 22-32: In the first paragraph of the conclusions the authors are very generic about their findings regarding the eddy bgc composition discussed in the sections from 3.1 to 3.4. Some of the results are not recalled (eg, particle, POM and DOM distribution). Since many interesting findings are discussed in the paper, I strongly suggest strengthening this part of the conclusion.

**AC37:** We agree and also see the need to improve this section. We will extend this section and ensure to cover all relevant findings of this paper.

**RC38**: Figure 2: colorbar and legend seem to contradict each other: if orange points refer to ACME (M105), what do the blue points refer to? This choice of colormap is unhelpful, the color is mostly constant.

**AC38:** We agree that the legend is misleading due to the color. We will change this in a revised version of this figure and may also consider adjusting the non-linearity of the colorbar.

**References:**

Fischer, G., Karstensen, J., Romero, O., Baumann, K.-H., Donner, B., Hefter, J., Mollenhauer, G., Iversen, M., Fiedler, B., Monteiro, I. and Körtzinger, A.: Bathypelagic particle flux signatures from a suboxic eddy in the oligotrophic tropical North Atlantic: production, sedimentation and preservation, Biogeosciences Discuss., 12(21), 18253–18313, doi:10.5194/bgd-12-18253-2015, 2015.

Hauss, H., Christiansen, S., Schütte, F., Kiko, R., Edvam Lima, M., Rodrigues, E., Karstensen, J., Löscher, C. R., Körtzinger, A. and Fiedler, B.: Dead zone or oasis in the open ocean? Zooplankton distribution and migration in low-oxygen modewater eddies, Biogeosciences, 13(6), 1977–1989, doi:10.5194/bg-13-1977-2016, 2016.

Karstensen, J., Schütte, F., Pietri, A., Krahmann, G., Fiedler, B., Grundle, D., Hauss, H., Körtzinger, A., Löscher, C. R., Testor, P., Vieira, N. and Visbeck, M.: Upwelling and isolation in oxygen-depleted anticyclonic modewater eddies and implications for nitrate cycling, Biogeosciences Discuss., 1–25, doi:10.5194/bg-2016-34, 2016.

Löscher, C. R., Fischer, M. A., Neulinger, S. C., Fiedler, B., Philippi, M., Schütte, F., Singh, A., Hauss, H., Karstensen, J., Körtzinger, A., Künzel, S. and Schmitz, R. A.: Hidden biosphere in an oxygen-deficient Atlantic open ocean eddy: future implications of ocean deoxygenation on primary production in the eastern tropical North Atlantic, Biogeosciences Discuss., 12(16), 14175–14213, doi:10.5194/bgd-12-14175-2015, 2015.

Schulz, K. G., Bellerby, R. G. J., Brussaard, C. P. D., Büdenbender, J., Czerny, J., Engel, A., Fischer, M., Koch-Klavsen, S., Krug, S. A., Lischka, S., Ludwig, A., Meyerhöfer, M., Nondal, G., Silyakova, A., Stuhr, A. and Riebesell, U.: Temporal biomass dynamics of an Arctic plankton bloom in response to increasing levels of atmospheric carbon dioxide, Biogeosciences, 10(1), 161–180, doi:10.5194/bg-10-161-2013, 2013.

Schütte, F., Brandt, P. and Karstensen, J.: Occurrence and characteristics of mesoscale eddies in the tropical northeastern Atlantic Ocean, Ocean Sci., 12(3), 663–685, doi:10.5194/os-12-663-2016, 2016.

Sheen, K. L., Brearley, J. A., Naveira Garabato, A. C., Smeed, D. A., Laurent, L. St., Meredith, M. P., Thurnherr, A. M. and Waterman, S. N.: Modification of turbulent dissipation rates by a deep Southern Ocean eddy, Geophys. Res. Lett., 42(9), 3450–3457, doi:10.1002/2015GL063216, 2015.

Thomsen, S., Kanzow, T., Krahmann, G., Greatbatch, R. J., Dengler, M. and Lavik, G.: The formation of a subsurface anticyclonic eddy in the Peru-Chile Undercurrent and its impact on the near-coastal salinity, oxygen, and nutrient distributions, J. Geophys. Res. Ocean., 121(1), 476–501, doi:10.1002/2015JC010878, 2016.

---

## Author Response (AR2)

Dear Reviewer, dear Editor

We very much appreciate that you took the time to also review the revised manuscript. We addressed all your comments and did minor modifications on the manuscript as suggested. We also double-checked the entire manuscript for typos again.

In the following we sorted all comments/questions (**RC:**) by numbering these (according to your numbering) and providing for each an answer (**AC:**).

Best regards,
B. Fiedler & Coauthors

**1. Reviewer #2:**

Review of Fiedler et al., Oxygen Utilization and Downward Carbon Flux in an Oxygen-Depleted Eddy inb the Eastern Tropical North Atlantic.
bg-2016-23
Minor Revision

**Specific comments:**

**Reviewer Comment (RC)1_:** *Sporadic upwelling (AC 1.4, General Comments, Major revision):*

*I appreciate the detailed explanation given by the Authors in their comment AC 1.4 regarding the submesoscale upwelling in the eddy. However, in Section 3.2 of the revised paper I haven't found the same explanation as clear as it is in the extended Authors Comment 1.4, since the summary-sentence referring to the paper by Karstensen et al. (2016) is confusing.*

*Page 12 lines 26-29: "As such, this finding is interpreted as being a signature of a vertical flux event related to submesoscale processes and stratification, which on the one side isolate the core and prevent oxygen supply while in parallel support vertical nutrient flux at the eddy rim (Karstensen et al., 2016)."*

*It is not clear what "vertical flux event related to submesoscale processes and stratification" means, and it is not clear who "isolate the core and prevent oxygen supply" and who "support vertical nutrient flux": "isolate", "prevent" and "support" refer to a plural subject but cannot be done interchangeably by both stratification and upwelling.*

*In my opinion this is a crucial passage for understanding how the biological activity in mixed layer of the ACME is sustained and how this can be reconciled with the isolation of the core. Therefore, I suggest to rephrase the above-mentioned sentence to distinguish between stratification and submesoscale vertical fluxes in a clear way, highlighting their contrasting but simultaneous work.*

*If my understanding is correct, on one side there is stratification that isolates the core and prevents oxygen supply in the core and on the other side there are submesoscale sporadic upwelling events at the rim of the eddy that feed the mixed layer with nutrients, justifying the combination of tracer concentrations measured above the core. I have not been able to find a final version of the cited paper by Karstensen et al. since it seems to be currently in discussion on Biogeosciences with major revisions requested, therefore I can only try to understand its conclusions at the present stage. I kindly ask the Authors to point me to the final manuscript if available.*

*The following is a quick attempt to rephrase the sentence and should be considered by the Authors as a mere suggestion: "The combination of tracers concentrations measured in the mixed layer of the eddy is interpreted as the signature of a submesoscale vertical flux event. On one side stratification isolates the core and prevents oxygen supply, on the other side submesoscale upwelling at the eddy rim supports the vertical nutrient flux in the mixed layer (Karstensen et al., 2016)."*

*The fact that the upwelling is supposed to happen at the rim of the eddy could also be mentioned in the Conclusions section (page 18 line 27), Eg. "…are caused by upwelling events at the rim of the eddy…".*

**Author Comment (AC)1_:** We fully agree with the reviewer that this particular sentence was not well formulated and rather confusing. We split this sentence in order to better separate these two processes from each other.

"The combination of nutrients concentrations measured in the mixed layer of the eddy is interpreted as the signature of a submesoscale vertical flux event. On one side stratification isolates the core and prevents oxygen supply, on the other side submesoscale upwelling at the eddy rim supports the vertical nutrient flux into the mixed layer of the eddy (Karstensen et al., 2016)."

We also added the information about upwelling occurring at the rim to section 2.4 and to the conclusion section:

"The high productivity is proposed to be driven by vertical nutrient flux at the rim of the eddy into the euphotic zone, a situation that resembles coastal upwelling regions."

"There is evidence that moderately elevated nutrient concentrations in the top layer of the ACME are caused by upwelling events at the rim of the eddy and fuel an enhanced surface primary productivity that moves with the ACME."

Regarding the Karstensen et al. paper this is being revised at the moment.

**RC2_:** *Eddy vertical structure (AC 15, Specific Comments)*
*I suggest that the Authors specify also the depth of the lower bound of the eddy core.*
*This could fit for example after page 11, lines 31-32 where the mixed layer depth is specified, and would help the reader to interpret the figures. Alternatively, horizontal dashed lines could be added to the plots (Figures 3,4,5,6) to highlight the depth of the core region.*

**AC2_:** This information was already provided in the text but have to admit that it was a bit hidden. We rephrased a sentence in order to emphasize this information in the text.
"This finding supports the isolation hypothesis for the eddy core as well as the assumed origin on the Mauritanian shelf of this particular eddy. The lower bound of the eddy core was found at approx. 250 m ($\sigma_\theta$=26.6 kg m$^{-3}$-1000) from where TS characteristics start to become more variable and no indication for isolation is found anymore."

**Technical comments:**

**RC1**: *page 3 line 21: "about 2 to 3 ACME were generated each year" - I suggest either to add in which years or to use "are generated" as a general statement*
**AC1**: Sentence edited. This was meant to be a general statement.

**RC2**: *page 4 line 27: "we did not had" should be "we did not have"*
**AC2**: Sentence corrected.

**RC3**: *page 4 line 30: "opportunistic Argo float data" - "opportunistic" is not the right adjective, probably "suitable" or "appropriate"*
**AC3**: We have replaced "opportunistic" by the word "appropriate".

**RC4**: *page 5 line 4: "the marine carbonate system functioning on low-oxygen eddies" - "on" should be "in"*
**AC4**: Sentence corrected.

**RC5**: *page 5 line 27: "This was also conducted outside of the eddy" - given the previous sentence, it's not clear what "This" refers to, maybe better to start with "CTD casts were also performed outside of the eddy"*
**AC5**: Sentence rephrased as proposed.

**RC6**: *page 6 lines 1-2: "This points out that these stations were more at the rim of the eddy, rather than in the surrounding water representing typical background conditions." - I find this sentence confusing, I would rephrase it, for example: "This points out that these stations were more at the*

*rim of the eddy rather than in the surrounding water, therefore they do not represent typical background conditions."*

**AC6**: Sentence rephrased as follows: "This points out that these stations were more at the rim of the eddy rather than in the surrounding water, therefore they may not represent typical background conditions."

**RC7**: *page 8 lines 28-30, page 9 line 1: "At the same time, subsequent sinking of particulate matter combined with an efficient isolation of the core from surrounding waters hinders oxygen ventilation." - I don't understand this statement: how does the sinking of particulate matter hinder oxygen ventilation? Is "hinders" (singular) only referring to the "efficient isolation"? Maybe this could be rephrased to better express how the sinking and the isolation contribute to lower the oxygen in the eddy core.*

**AC7**: We agree that the causality within this sentence was wrong. We edited this section as follows: "Karstensen et al. (2015) suggested that the low-oxygen cores of the eddies were created by an enhanced subsurface respiration due to high surface productivity. Subsequent sinking of particulate matter produced in the surface layer fuels this process. At the same time, an efficient isolation of the core from surrounding waters hinders oxygen ventilation."

**RC8**: *page 9 line 27: Data is a plural noun, "was" should be "were"*
**AC8**: Sentence corrected.

**RC9**: *page 11 line 22: "westward propagation from the shelf into the open." - This sentence misses a final noun, eg. "ocean" or "sea" or "waters"*
**AC9**: Apparently, this word was cut off during the former revision. Final noun added: "This underlines the isolation of the eddy against mixing processes with surrounding waters during its westward propagation from the shelf into the open ocean."

**RC10**: *page 11 line 22: "being" is not needed: This hypothesis is further corroborated […]*
**AC10**: We removed the word "being" from this sentence.

**RC11**: *page 11 lines 26-27: numbers in brackets should be accompanied with the unit (years)*
**AC11**: Units were added to the numbers in brackets.

**RC12**: *page 12 line 12: when talking about the decrease "of about 57.0 umol/kg to suboxic levels", an initial shelf value and a precise final value should be specified in the text "of about 57.0 umol/kg, from X to Y"*
**AC12**: Thank you for this remark. We have revised this sentence by adding the requested information. We also decided to not state the maximum difference along the depth horizon, but rather give the number along the respective isopycnal. This is also used for OUR calculations later in the manuscript (see. Also table 1). Sentence edited as follows: "In comparison to the reference profile from the Mauritanian Shelf, we find a maximum oxygen decrease in the eddy core at a depth of 100 m ($\sigma\theta$ = 26.35 kg m$^{-3}$-1000) of up to 44.4 µmol kg$^{-1}$, from 48.9 µmol kg$^{-1}$ at to shelf to 4.5 µmol kg$^{-1}$ during M105 (Figure 3; Table 1 )."

**RC13**: *page 12 lines 22-23: "associated with the passage of a former ACME passing the observatory" - passing should be substituted, eg. "near", "in the region of", "in correspondence of"*
**AC13**: Word replaced by "near".

**RC14**: *page 13 line 14: "This value is very much in contrast to the regional background condition" - This sentence should be rephrased, eg. "This value is in clear contrast with the regional background condition"*

**AC14**: Sentence rephrased: "This value is in clear contrast with the regional background condition at CVOO, where $\Omega_{Ar}$=1 is found below 2500 m depth and the typical $\Omega_{Ar}$ at 100 m depth is approx. 2.4.".

**RC15**: *page 17 line 23: missing "those": "…show higher values than those found…"*
**AC15**: Word added to the sentence.

**RC16**: *page 18 line 25: "continuing" should be substituted with "continuous" or "persistent"*
**AC16**: Word substituted.

[revised manuscript text omitted]